# Char dominates black carbon aerosol emission and its historic reduction in China

Junjie Cai [1,8], Hongxing Jiang [1,8], Yingjun Chen [1,2] ✉, Zeyu Liu [1], Yong Han[1], Huizhong Shen [3], Jianzhong Song [4,5], Jun Li [4,5], Yanlin Zhang [6,7], Rong Wang [1], Jianmin Chen [1] & Gan Zhang [4,5] ✉

Emission factors and inventories of black carbon (BC) aerosols are crucial for estimating their adverse atmospheric effect. However, it is imperative to separate BC emissions into char and soot subgroups due to their significantly different physicochemical properties and potential effects. Here, we present a substantial dataset of char and soot emission factors derived from field and laboratory measurements. Based on the latest results of the char-to-soot ratio, we further reconstructed the emission inventories of char and soot for the years 1960–2017 in China. Our findings indicate that char dominates annual BC emissions and its huge historical reduction, which can be attributable to the rapid changes in energy structure, combustion technology and emission standards in recent decades. Our results suggest that further BC emission reductions in both China and the world should focus on char, which mainly derives from lower-temperature combustion and is easier to decrease compared to soot.

Black carbon (BC) aerosol, derived from the incomplete combustion of biomass and fossil fuels[1,2], is of major concern due to its adverse effects on climate[2–5], environment[6–8] and human health[9–15]. The short-lived climate pollutant has a positive radiative forcing on the climate system, ranking second only to carbon dioxide and methane[2,5]. The dome effect of BC lowers the development of the planetary boundary layers, leading to higher instances of extreme haze pollution[6,7] and contributing to the spread of wildfires through a complex feedback process[8]. Studies have confirmed that exposure to BC aerosol can lead to various health issues[10–13], including lower birth weight[11] and heart attacks[14]. To accurately estimate the impact of BC, emission factors of BC for various sectors have been measured and bottom-up emission inventories have been developed at national and global scales in recent decades[16–18].

It is imperative to differentiate between char and soot in current BC emissions as they are distinct materials with different physical structure[19,20], optical properties[21,22] and potential bio-toxicity[14], although they are originally separated by different temperatures in the Interagency Monitoring of Protected Visual Environments (IMPROVE) protocol and the thermal-optical reflectance (TOR) method for organic and elemental carbon analysis[23,24]. Therefore, treating char and soot as a unified entity may significantly skew the estimation of BC's various effects. However, separating BC into char and soot subgroups needs specific and reliable values of char to soot ratios for different emission sectors, resulting from differences in fuel composition, combustion condition, and after-treatment technologies[25]. For instance, biomass burning produces BC aerosol with a char fraction of

[1]Shanghai Key Laboratory of Atmospheric Particle Pollution and Prevention (LAP3), Department of Environmental Science and Engineering, Fudan University, Shanghai 200438, China. [2]Shanghai Institute of Pollution Control and Ecological Security, Shanghai 200092, China. [3]Guangdong Provincial Observation and Research Station for Coastal Atmosphere and Climate of the Greater Bay Area, School of Environmental Science and Engineering, Southern University of Science and Technology, Shenzhen 518055, China. [4]State Key Laboratory of Organic Geochemistry, Guangzhou Institute of Geochemistry, Chinese Academy of Sciences, Guangzhou 510640, China. [5]Guangdong-Hong Kong-Macao Joint Laboratory for Environmental Pollution and Control, Guangzhou Institute of Geochemistry, Chinese Academy of Science, Guangzhou 510640, China. [6]Yale–NUIST Center on Atmospheric Environment, International Joint Laboratory on Climate and Environment Change (ILCEC), Nanjing University of Information Science and Technology, Nanjing 210044, China. [7]Jiangsu Provincial Key Laboratory of Agricultural Meteorology, College of Applied Meteorology, Nanjing University of Information Science and Technology, Nanjing 210044, China. [8]These authors contributed equally: Junjie Cai, Hongxing Jiang. ✉e-mail: yjchenfd@fudan.edu.cn; zhanggan@gig.ac.cn

over 80%[21], whereas the contribution of char and soot to BC is almost equal in vehicle exhaust[26].

With the rapid development of the economy, China became the world's largest emitter of BC in 2000, accounting for about one-fourth of the global BC emissions[27]. However, with the update of air pollution control policies and optimization of combustion technologies, China's BC emissions decreased by about one-third by 2019. The rapid change in China's BC emissions holds significant reference value for other developing countries[28]. Multifarious BC emission sectors can be observed in China with different combinations of fuel composition and combustion conditions, including residential stove combustion (RS) of solid fuels such as biomass and coal, which occurs in rural or undeveloped mountainous areas for heating and cooking, non-road and on-road vehicle exhaust (VE), and coal-fired boiler combustion (CB) in industrial and power plants, with corresponding emission standards gradually converging with or partly stricter than those in the developed countries. In recent decades, China has undergone significant socio-economic development, leading to a shift in energy structure and advancements in combustion and emission control technologies. As a result, there has been a significant reduction in BC emissions from 2609 Gg in 1995 to 1229 Gg in 2017[29]. China's rapid reduction of BC emissions provides a valuable reference for other countries, in which the contribution of char and soot subgroups is an important topic, especially in relation to different emission sectors or combinations of fuel types, combustion conditions and control measures.

In this study, we collected combustion event data from three different sectors, including 122 from RS, 160 from VE, and 12 from CB (specific information shown in Table 1 and Text S1). Based on this data, we established a comprehensive multi-sector database of char and soot emission factors. By further combining 126 sets of results from laboratory stove simulations, we identified key factors influencing the variation in char and soot emissions and explained the differences in their formation pathways. Additionally, we have reconstructed historical emission inventories of char and soot separately in China to better understand the driving forces behind BC emission reductions. Based on our findings, we have made recommendations for future BC reduction efforts on a global scale.

## Results and discussion

### Char contributes higher proportion of BC in emission sources with higher BC emission factors

A total of 420 samples were collected and measured, including residential biomass burning (RBB) and residential coal combustion (RCC) in the RS sector, on-road vehicle exhaust (OVE) and non-road vehicle exhaust (NVE) in the VE sector, and coal-fired boilers for industry (ICB) and power plants (PCB) in the CB sector. These sectors represent the major combustion sources in China's BC emission inventory[17] (Table S1).

Field measurements revealed that the residential stove combustion sector had the highest emission factors (EFs) of BC ($EF_{BC}$), with RBB at $1.00 \pm 0.64$ g/kg and RCC at $0.60 \pm 0.23$ g/kg. The vehicle exhaust sector followed with NVE at $0.34 \pm 0.18$ g/kg and OVE at $0.05 \pm 0.03$ g/kg, while the coalfired boiler sector (ICB at $0.86 \pm 0.82$ mg/kg and PCB at $0.11 \pm 0.12$ mg/kg) was lower by three orders of magnitude (Fig. 1a). This variation of $EF_{BC}$ among emission sectors could be jointly explained by the fuel composition (e.g., the difference between solid fuels and petroleum products), combustion technology (e.g., coal combustion in stoves versus boilers) and emission standards (e.g., the difference between NVE and OVE). Compared to the $EF_{BC}$ data used in the historical inventory (Fig. S1), we observed the following differences: For residential stove combustion, the $EF_{BC}$ for RBB has not significantly changed, while the $EF_{BC}$ for RCC has decreased by about 50% due to continuous improvements in stove technology. The $EF_{BC}$ for vehicle exhaust has decreased by one to two orders of magnitude due to updated emission standards and fuel quality. For coalfired boiler, especially PCB, the $EF_{BC}$ has decreased by 3–4 orders of magnitude, attributed to the closure of residential coke ovens and the continuous strengthening of emission standards. Moreover, the trend of the $EF_{BC}$ in our results is consistent with the data found in the literature within the past 5 years (Table S2). Therefore, $EF_{BC}$ data used for emission inventory calculation needs to be updated in a timely manner.

The emission factors measured for char ($EF_{char}$) and soot ($EF_{soot}$) were presented in Fig. 1b, c. $EF_{char}$ demonstrated a similar pattern to $EF_{BC}$ across various emission sources. RBB ($0.81 \pm 0.56$ g/kg) and RCC

## Table 1 | Sample information and collection methods from different emission sectors

| Sector/Type | Materials and categories | Stoves and conditions | Total Sample numbers |
|---|---|---|---|
| Residential Stove Combustion (RS) | | | |
| Residential Biomass Burning (RBB) | Nine crop straws (maize, corncob, soybean, rice, cotton, sorghum, reed, peanut, and bamboo), log-wood and brushwood | Residential brick biomass stoves | 84 |
| Residential Coal Combustion (RCC) | Two coal types (Chunk and Briquette), | One for honeycomb briquettes and three traditional style chunk stoves | 38 |
| Vehicle exhaust (VE) | | | |
| On-road vehicle exhaust (OVE) | Fifteen On-road vehicles with different oil type (Gasoline: 93#; Diesel: 0#;) four different emission stand (China III- China VI) | Idling; suburban main road driving; high speed road driving | 63 |
| Non-road vehicle exhaust (NVE) | Six forklifts with or without DPF | Under different working mode (idling, unloaded walking and loaded walking) | 97 |
| | Three oceangoing vessels (OGV) Steady-state sailing conditions, | Engine load (25%, 50%, 75%, 100%), engine rated power, and fuel type (marine gas oil and heavy fuel oil). | |
| Coal-fired boiler combustion (CB) | | | |
| Coal-fired boilers for industry (ICB) | Two drugs manufacturing factory, coal-fired boilers and two coking plants (ICBs)) in Hebi | | 9 |
| Coal-fired boilers for power plants (PCB) | Coal-fired power plant, biomass power plants (PCBs) | | 3 |
| Simulated combustion in laboratory | | | |
| Residential solid fuel combustion in quartz tube furnace (RS-QTF) | Five types of biomasses (rice straw, wheat straw, corn stalks, pine wood, and poplar wood) | Combusted at 300-900°C, (total seven temperature levels) with oxygen concentration of 21% and 10.5% in the combustion furnace. | 126 |
| | One type of coal (Xuzhou coal) | | |

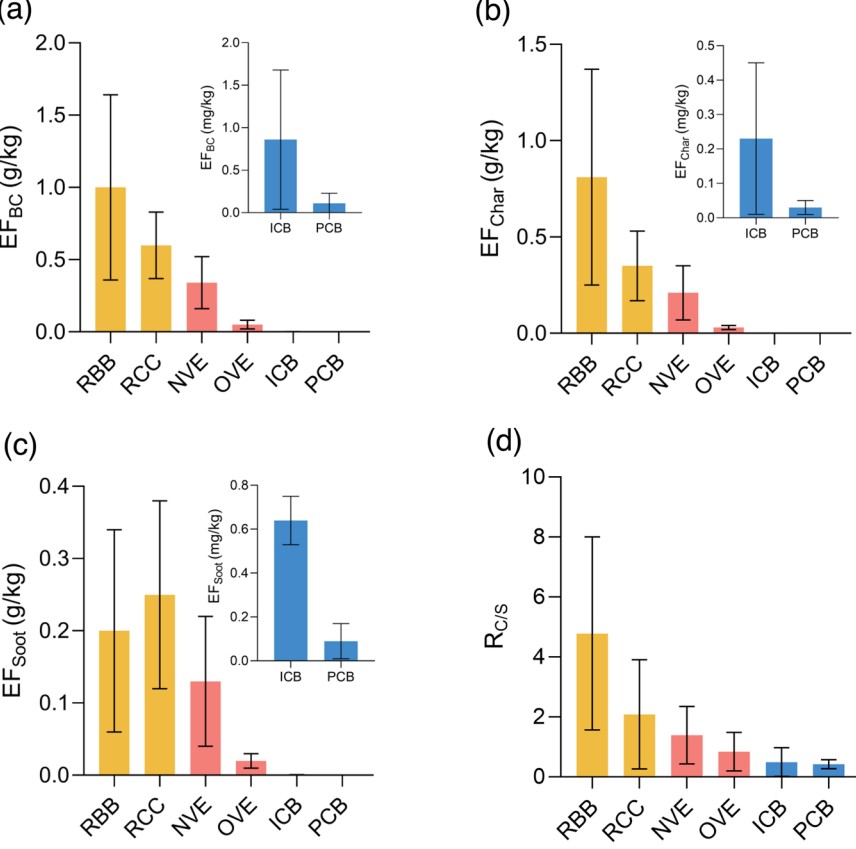

**Fig. 1 | Emission characteristics and proportion of black carbon (BC), char, and soot across different emission sources.** Emission factors (EF) of BC (**a**), Char (**b**), Soot (**c**) and the ratio of char to soot ($R_{C/S}$, **d**) for different emission sources in three emission sectors (Yellow refers to residential biomass burning (RBB) and residential coal combustion (RCC) in the residential stove combustion (RS) sector, red refers to on-road vehicle exhaust (OVE) and non-road vehicle exhaust (NVE) in the vehicle exhaust (VE) sector, blue refers t0 coal-fired boilers for industry (ICB) and power plants (PCB) in the Coal-fired boiler combustion (CB) sector. The error bar refers to the standard deviation of each sources).

$(0.35 \pm 0.18\,\text{g/kg})$ had notably higher values than NVE $(0.21 \pm 0.14\,\text{g/kg})$ and OVE $(0.03 \pm 0.01\,\text{g/kg})$, as well as ICB $(0.22 \pm 0.23\,\text{mg/kg})$ and PCB $(0.03 \pm 0.03\,\text{mg/kg})$ by three orders of magnitude. On the other hand, the trend of $EF_{soot}$ was slightly different. RCC $(0.25 \pm 0.13\,\text{g/kg})$ was comparable to RBB $(0.20 \pm 0.14\,\text{g/kg})$ and NVE $(0.13 \pm 0.09\,\text{g/kg})$, followed by OVE $(0.02 \pm 0.01\,\text{g/kg})$, ICB $(0.64 \pm 0.11\,\text{mg/kg})$, and PCB $(0.09 \pm 0.08\,\text{mg/kg})$. It is evident that char plays a significant role in $EF_{BC}$ of RS and VE sectors. Notably, char contributed exceeded 80% in $EF_{BC}$ for RBB and approximately 60% for RCC, NVE, and OVE while soot contributed over 70% for high-temperature combustion sources such as ICB and PCB. This suggests that char and soot may have varying sensitivities to fuel and combustion conditions, as well as distinct formation pathways, which will be further discussed later.

The ratio of $EF_{char}$ to $EF_{soot}$ ($R_{C/S}$) was calculated to gain a more intuitive view on the contribution of these two subgroups to the $EF_{BC}$ across various sectors. $R_{C/S}$ is a reliable indicator for distinguishing among different sectors. As presented in Fig. 1d, the values of $R_{C/S}$ were found to be higher than 2.00 for RS (RBB $4.78 \pm 3.22$, RCC $2.08 \pm 1.82$), slightly higher than 1.00 for VE (NVE $1.39 \pm 0.60$, OVE $1.14 \pm 0.92$), and lower than 0.50 for CB (ICB $0.49 \pm 0.50$, PCB $0.42 \pm 0.15$). Thus, char contributes a higher proportion of BC emissions in sources with higher $EF_{BC}$. The wide range of $R_{C/S}$ values observed among various sources, which is consistent with previous studies (Table S3), confirms that the values are highly sensitive to fuel composition, combustion conditions, and after-treatment technology. It also suggests that the actual proportion of char and soot in $EF_{BC}$ for each source should be taken into account when calculating the dividing BC emission inventory into char and soot subgroups.

## Char dominates annual BC emissions and its huge historical decrease in China

The historical inventory of BC emissions in China from 1960 to 2017 has been recently updated[17] (Fig. 2a). By combining the latest BC emission inventories with the $R_{C/S}$ values of emission sources measured in our study, we were able to separate and reconstruct the historical emission inventories of char and soot during the same period. This allowed us to determine the contribution of char and soot to annual BC emissions in China (Fig. 2b and Table S4). The calculation is based on the assumption that there is no significant change between historical and current $R_{C/S}$ for major emission sources, with the exception of beehive coke ovens, an important industrial source that existed mainly in 1990-2000 with combustion technologies and after-treatment measures much closer to RCC than the current ICB[29]. Therefore, we set the $R_{C/S}$ value of 1.29 for this particular source by averaging RCC $(2.08 \pm 1.82)$ and ICB $(0.49 \pm 0.50)$ during this period. As shown in Fig. 2, the history of BC emissions from 1960–2017 can be divided into three distinct periods: a rapid decline before 1989 (PI), a dramatic fluctuation between 1990 and 2000 (PII), and a slow decline after 2001 (PIII).

During the PI period, annual BC emissions increased from 1540 Gg in 1960 to 2056 Gg in 1989, with a minimum of 1264 Gg in 1962. Both char and soot emissions displayed similar upward trends, with annual char emissions rising from 1054 Gg in 1960 to 1206 Gg in 1989, and soot emissions increasing from 486 Gg to 851 Gg. However, the ratio of annual char to soot emissions (named $R_{CE/SE}$) showed a downward trend over time on a relatively high level, ranging from $2.17 \pm 0.80$ in 1960 to $1.42 \pm 0.42$ in 1989, with a maximum of $3.28 \pm 1.47$ in 1963. The

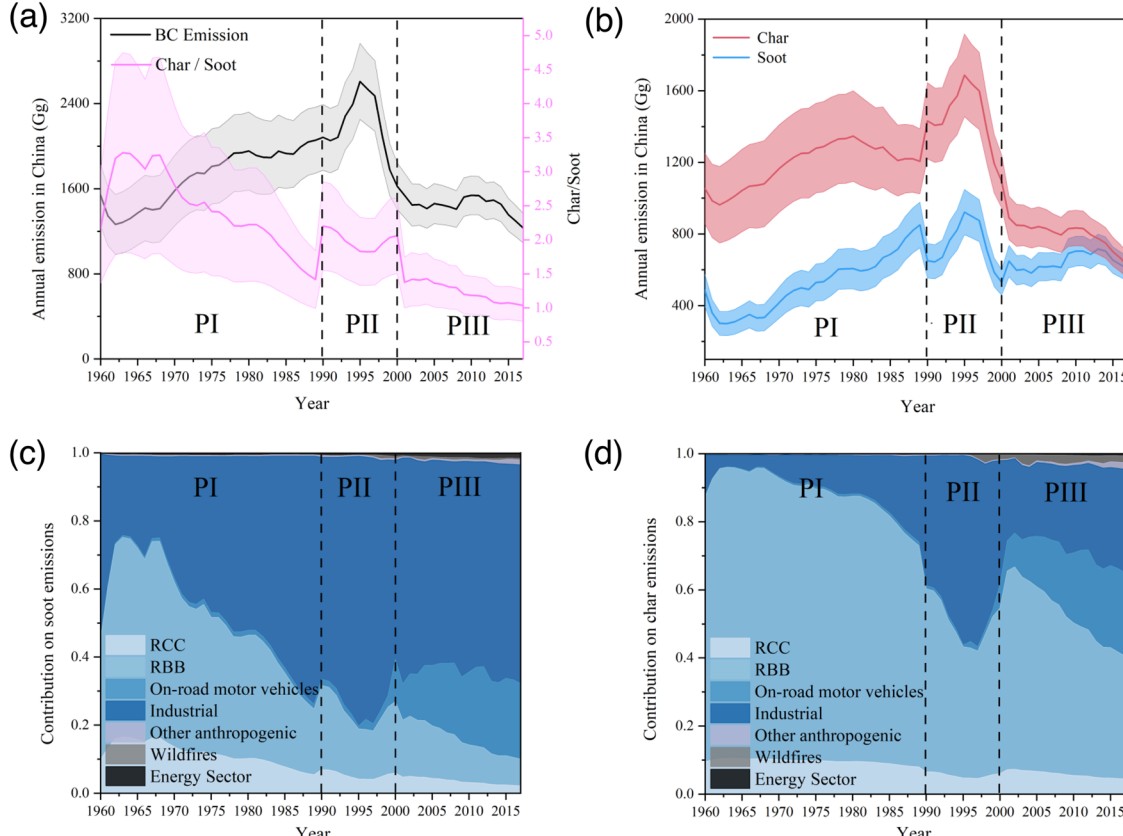

**Fig. 2 | Historical emissions and source contributions of black carbon (BC), Char, and Soot in China from 1960–2017.** Total BC and char to soot ratio of anthropogenic source emissions in China from 1960–2017 (**a**), temporal trends of annual emission of char and soot (**b**) (Shaded areas are uncertainty intervals of char and soot, represented by quartiles). The percentage contribution of each emission source to char (**c**) and soot (**d**) (RBB refers to residential biomass burning, RCC refers to residential coal combustion, Other anthropogenic refers to emissions from non-road transportation, Energy sector refers to emissions from power plants. PI–PIII refers to three periods we defined in the manuscript).

emission source profile of BC during this period resembles that of most developing countries today, with residential solid fuel burning (RBB and RCC) contributing 51–91% of BC emissions, while the contributions of transportation and industrial sources are much lower (Fig. 2c, d).

During the PII period, annual BC emissions experienced a rapid rise from 2083 Gg in 1990 to 2609 Gg in 1995, followed by a sharp decline to 1623 Gg in 2000. Similarly, emissions of char and soot peaked at 1686 Gg and 922 Gg in 1995, respectively, before declining rapidly to 1093 Gg and 530 Gg in 2000. However, the $R_{CE/SE}$ remained relatively stable throughout the period, ranging from 2.20 ± 0.65 in 1990 to 2.06 ± 0.56 in 2000, with a minimum of 1.82 ± 0.49 in 1996. The emission trends for BC and its subgroups are consistent with the proliferation of beehive coke ovens in China prior to 1995, and their eventual phase-out following the enactment of the Coal Law in 1996[29]. The contribution of this source to BC emissions rapidly increased to approximately 55% for char and 79% for soot during 1990–1995 (Fig. 2c, d), surpassing the RS sector, before declining to around 20% for char and 60% for soot during 1996–2000.

During the PIII period, annual BC emissions saw a gradual decline from 1538 Gg in 2001 to 1229 Gg in 2017, representing a reduction of approximately 20%. However, the trends for char and soot emissions differed. Char emissions experienced a rapid decrease from 890 Gg to 625 Gg, reflecting a decline of nearly 40%, whereas soot emissions only slightly dropped from 648 Gg to 604 Gg, indicating a decrease of less than 7%. The $R_{CE/SE}$ during this period showed a continuous decrease, albeit at a lower level than during the PI period, from 1.37 ± 0.37 in 2001

to 1.04 ± 0.23 in 2017. Huang et al. recently reported an average $R_{CE/SE}$ value of 1.41 ± 0.71 for particulate BC in surface water samples from Taihu Lake, the third largest freshwater lake in China, which was mainly derived from atmospheric deposition and reflected the BC emission composition of surrounding urban cities in 2019[30]. The continuously increasing contributions to BC emissions were observed for CB (from 41% to 47%) and VE (from 9% to 23%), whereas the contribution from the RS sector decreased from 47% to 25% during the PIII period. This is consistent with the rapid development of the industrial and vehicular sectors[31–33], the shift towards modern energy sources (such as gas and electricity)[34], and the improvement of residential stove technology (including the installation of second combustion chambers)[34,35]. As a result, the BC emission source profile gradually approached that of developed countries during this period[17].

In general, it can be concluded that char dominates the annual BC emissions and its huge historical reduction in China. The average ratio of annual $R_{CE/SE}$ during 1960–2017 was 2.01, especially with the average $R_{CE/SE}$ value of 2.33 in the PI and PII periods, indicating that char contributed around 70% of the annual BC emissions in the history (Fig. 2b). On the other hand, the condition in the PIII period (i.e., 2001–2017) is somewhat different from the previous two periods, with the average $R_{CE/SE}$ decreasing to 1.23. Especially after 2013, when the Chinese government promulgated the toughest-ever Air Pollution Prevention and Control Action Plan, the $R_{CE/SE}$ value further decreased to 1.10, making the contributions of char and soot to BC emissions almost identical[36]. In the BC emission reduction of 309 Gg during 2001–2017, char emission decreased by 265 Gg and contributed 86% of the BC reduction, while soot decreased by 44 Gg and contributed only 14% of

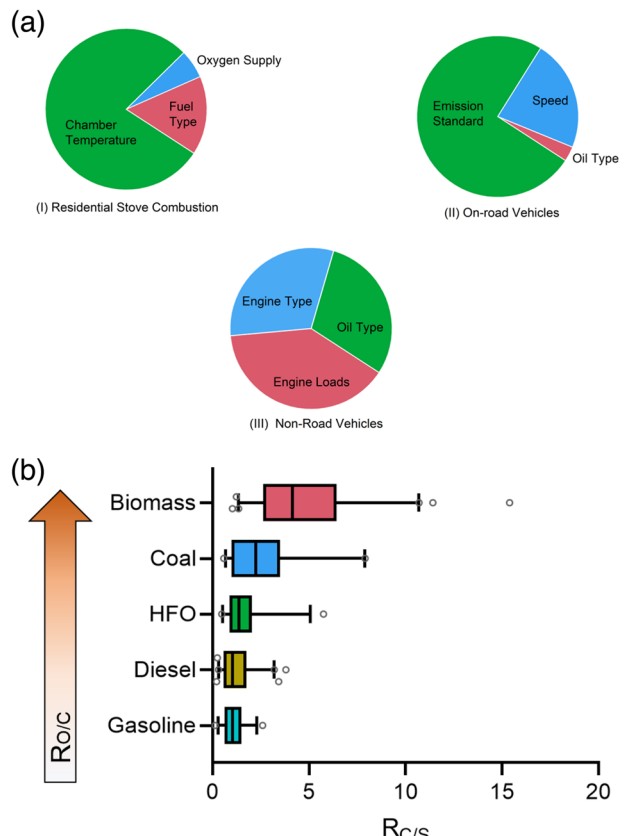

**Fig. 3 | The impact of different factors on the ratio of char to soot ($R_{C/S}$).** **a** Proportional contribution of different influencing factors to the $R_{C/S}$ variations in residential stove combustion (I), on-road vehicle exhaust (II) and non-road vehicle exhaust (III). **b** Effects of oxygen to carbon ratio ($R_{O/C}$) in fuel composition on $R_{C/S}$.

BC reduction in this period. This highlights the dominance of char in China's historical BC emission reduction.

The historical trends of atmospheric BC emissions and the proportion of char to soot are confirmed by their concentrations recorded in several sediment cores around China. These cores, including Taihu Lake and Chaohu Lake in Eastern China, Nam-Co and Dagzo-Co Lake in the Tibetan area, Dianchi Lake in Southeastern China, and the East China Margin Sea[37–42], showed a continuous increase in BC and char concentrations starting from the 1960s, peaking around 1995, and then declining significantly[40–42]. It is important to note that there are differences in the treatment of sediment and aerosol samples[43,44], as well as regional socioeconomic characteristics, which can affect the results of these sediment core samples. Nonetheless, the historical ratios of char concentration to soot in East China Sea sediment cores also show a similar trend to the annual emission ratio of char to soot, with an increasing trend since 1960, followed by a significant peak in the 1990s[40] (Fig. S2).

## Different sensitivity and formation pathway between char and soot

As described above, there are significant differences in the EFs of char and soot, as well as their ratios, across various sources. Char accounts for a higher proportion of BC from the sources with higher EF$_{BC}$ (i.e., RBB, RCC, NVE). Therefore, it can be expected that there are notable differences in the sensitivity of char and soot to fuel/combustion combinations and even their formation pathways. The dominance analysis results (Fig. 3a) demonstrate that furnace temperature is the most significant influencing factor for lower-temperature combustion of solid fuels (RBB and RCC), explaining 79% of the variation in $R_{C/S}$

values, followed by fuel composition (18%). For NVE with relatively poor engine technology, fuel oil quality and emission standards, engine parameters (load and power) and oil composition explain 70% and 30% of the variation in $R_{C/S}$, respectively. In contrast, for sources with more advanced combustion technologies and more stringent emission standards, such as OVE, emission standards or after-treatment technologies explain 75% of the variation in $R_{C/S}$, which masks the difference in the effects of fuel composition and formation pathways of char and soot.

In the past few decades, through experimental, theoretical, and computational work, many pathways for the formation of soot have been proposed[45–47]. However, the specific mechanisms of char formation have not been clearly elucidated to date. Through controlled combustion experiments, we have observed limited information, revealing differences between char and coal smoke in terms of precursors and byproducts.

Firstly, compared to soot, char may be formed from higher molecular weight organic volatile precursors. We observed that the first stage of coal combustion produces smaller organic molecules, rapidly generating coal smoke, while the second stage of pyrolysis produces larger organic fragments or tar, accompanied by the rapid formation of char[19]. This phenomenon may also occur in internal combustion engines, where lower molecular weight components rapidly evaporate when the fuel mixture is injected into compressed hot air, resulting in the combustion of soot upon contact with oxygen, while higher molecular weight components form char due to limited contact with oxygen[48,49]. Our experimental data confirms this inference: under low engine load or idle mode, the $R_{C/S}$ is higher, while under high engine load or operational mode, the $R_{C/S}$ is lower (refer to Table S1).

Secondly, different by-products or intermediate products are generated between char and soot formation pathways. We found that the high time-resolved $R_{C/S}$ were closely proportional to higher ratios of high-molecular-weight (4–6 ring) to low-molecular-weight (2–3 ring) PAHs as well as higher ratios of refractory organic carbon to volatile organic carbon of OC fractions[50,51]. Furthermore, oxygenated polycyclic aromatic compounds (oPACs) are likely to be related to char production compared to parent PAHs in traditional soot formation process. This is supported by the fact that the oPACs content of RS samples (with high $R_{C/S}$) is three times higher than that of VE samples (with relatively low $R_{C/S}$) in our groups latest research. Since the oPACs are believed to derive from the conversion of oxygen-containing functional groups in the fuel during combustion, we can preliminarily conclude that that when fuels with relatively complex compositions (especially those with higher oxygen content) are burning at lower temperatures (Fig. 3b), more char than soot will be produced.

## Future BC reduction should pay more attention on char emission

As demonstrated, char is the primary contributor to annual BC emissions and has been the focus of the historical BC reduction efforts in China over the last few decades. Char is formed through lower-temperature combustion and can be controlled through optimized combustion conditions, making it a more manageable source of emissions than soot. Therefore, we recommend that future effort to reduce BC emissions in China and other countries, such as India, Nigeria, Congo, Indonesia[17], should prioritize on controlling sources with high char emissions. In these countries, significantly high $R_{C/S}$ values were often observed from urban ambient PM$_{2.5}$ samples (e.g., $R_{C/S}$ exceeded 2.0 in urban environmental samples in India[52–54]). Thus, there is a significant opportunity for reducing both BC and char through the adoption of advanced combustion and after-treatment technologies.

To be more precise, reducing emissions of BC and char in the RS and VE sectors is the primary focus for emission reduction efforts in

China. These sectors have significantly high emission factors for BC and R_{C/S}. Therefore, it should be recommended that solid fuels such as biofuel and coal be burned in boilers instead of small stoves, including advanced ones, in the RS sector. Boilers provide better combustion conditions and after-treatment measures, resulting in a reduction of EFs of BC and char by more than three orders of magnitude compared to residential stoves. However, advanced stoves have less impact on BC and char reduction compared to traditional ones (Fig. S3). In the VE sector, increasingly strict emission limits have shown significant benefits for reducing emissions of BC and other pollutants from on-road diesel trucks (e.g., EFs of BC and char from trucks with China-VI emission standard are lower by one order of magnitude than with China III, Table S1). These limits should be used as a reference for non-road mobile machinery burning diesel and heavy oil, such as agricultural and construction machines and ships. Currently, NVE is an important but neglected source of BC emissions, and its contribution is not considered in the BC emission inventory. According to the China Mobile Source Environmental Management Annual Report[55], PM emissions from NVE were 234 Gg in 2021, which is 3.4 times that of OVE. The lagging emission limits for NVE could largely explain the difference in EF_{BC} and EF_{Char} between NVE and OVE, followed by slightly backward fuel quality and engine technology for non-road mobile machinery. Implementing on-road mobile machinery emission limits will reduce BC and char emissions from non-road mobile machinery by about seven times compared to the current situation.

Therefore, if the above two measures are implemented—using solid fuels in boilers instead of stoves and enforcing OVE emission limits for NVE—BC emissions in both China and other countries will decrease significantly. In China, BC emissions will reduce by 572.27 Gg, accounting for 47% of the amount emitted in 2017[17]. Char will contribute to 68% of the BC reduction in China. In fact, not only in China, but also globally, the combustion of low-temperature solid fuels and emissions from mobile sources contribute significantly to BC. For example, in 2017, in India, another developing country, the contribution of mobile sources and the combustion of low-temperature solid fuels to BC emissions was approximately 66%, whereas globally, this proportion reached as high as 77%[17]. Considering that most countries around the world are relatively lagging behind China in terms of combustion technology and emission standards, promoting these two technological improvements globally will greatly facilitate the reduction of BC emissions, especially in reducing char emissions primarily generated from low-temperature combustion.

## Methods

### Sampling

In this study, we conducted field measurements on EFs of BC, char and soot for major emission sources (RBB, RCC, NVE, OVE, ICB, PCB), covering the source categories listed in BC emission inventories[16,17,56]. We also conducted laboratory experiments of solid fuel combustion in a tube furnace and on-board measurements of engine exhaust separately considering the engine type/oil quality/speed (for OVE) or engine type/load/oil/operation mode (for NVE), in order to further illustrate the influence mechanism and formation pathway of char and soot in RS and VE. The overall information of the field and laboratory measurements is shown in Table S5. Detailed information on fuels, dilution sampling methods, and samples can be found in Text S1 and in the references listed. A brief summary is as follows.

Flue gases emitted from emission source are sampled and measured by a portable dilution sampling system that includes a dilution channel, two flue gas analyzers (GA21plus, Madur, Austria), and sampling instruments (Fig. S4). A portion of the flue gas emitted from the stack is drawn into the dilution channel and mixed with clean air filtered with polypropylene fibers of 1 μm pore size for cooling and dilution. Pre- and post-dilution CO and $CO_2$ concentrations are

measured online by the two flue analyzers to obtain the actual dilution ratio which is 10–20 in the sampling. When selecting the dilution ratio, we took into account both the water vapor concentration and the sample concentration. Firstly, for engine exhaust from vehicles as well as industrial emissions, the water vapor concentration in the smoke is close to 100%, and water vapor condenses at a concentration of around 10%. Therefore, we chose the lowest dilution ratio of 10 to ensure that water vapor does not condense in the smoke pathway and to ensure that the collected sample quantity does not fall below the instrument's detection limit. Secondly, for household stoves with high particulate matter concentrations in the smoke, we chose a relatively higher dilution ratio of 20 to ensure that the sample does not exceed the loading capacity of the filter membrane. All samples were collected on pre-cleaned quartz filters. Samples were stored in a freezer at −20 °C under refrigeration pending analysis.

### Analyzing methods

Black carbon (quantified as BC using optical methods and as EC using thermal optical methods, these two concepts can be used interchangeably) is known as a mixture of compounds which corresponds to a carbon continuum, including char and soot subgroups. The concept of char and soot was first proposed by scientists in soil and their difference was observed qualitatively[44,57]. Han et al. first discriminated char and soot using IMPROVE protocol and thermal/optical reflectance (TOR) method, and found that the pure char materials always peaked at EC1 while pure soot samples peaked at EC2 and EC3 in TOR method[23].

In this study, char and soot of quartz-filter samples were measured using a thermal/optical carbon analyzer with the IMPROVE protocol and TOR method[51]. This protocol defines organic carbon (OC) and elemental carbon (EC) using two progressive heating stages with inert (pure He) and oxidizing (98% He/2% $O_2$) atmosphere, respectively, and further records OC and EC subgroups by heating temperatures, i.e., OC1 (140 °C), OC2 (280 °C), OC3 (480 °C), OC4 (580 °C), EC1 (580 °C), EC2 (740 °C), EC3 (840 °C); pyrolytic carbon (PyC) is a fraction of OC that is corrected by the reflectance laser signal and subtracted from EC. Thus, the actual OC and EC values are OC1 + OC2 + OC3 + OC4 + PyC and EC1 - PyC + EC2 + EC3, respectively. Among EC fractions, EC1-PyC and EC2 + EC3 are identified separately as char and soot[23,24].

### Data analysis and model performance

Emission factor (EF) for carbonaceous fractions from each combustion sources were calculated using carbon balance method[58–60]. The principle for carbon balance method and modified combustion efficiency (MCE) are shown in Text S1. Weighted average methods in this study are shown in Text S2. Data analysis was performed using Microsoft Excel and IBM SPSS Statistics 26. The relative contribution of each predictor variable (mean and standard deviation) was quantified using the Dominance Analysis package in R (v4.2.1, R Foundation for Statistical Computing, Vienna, Austria). In scientific research, an important issue is to determine the specific contribution of different explanatory variables to the variance of the dependent variable. In 2007, Israeli proposed a method called importance analysis based on previous studies. This method quantitatively determines the contribution of different explanatory variables to the determination coefficient ($r^2$) and variance of the dependent variable in linear regression, by inputting multiple explanatory variables and the dependent variable[61]. In recent years, this method has also been widely used in the environmental field to determine the potential impacts of various influencing factors on emissions[21,62,63]. The McFadden index ($r^2$.m) is referred to as the closer equivalent of the coefficient of determination in linear regression and is used as a parameter to express the relative importance of each independent variable in the dominance analysis.

The data on total BC emissions in China during 1960–2017 were obtained from the overall update of Chinese BC data by ref. 17. In this study, BC inventory is separated by substituting the measured char/soot of different emission sectors separately. Given the timeliness of the inventory, we did not update the $EF_{BC}$ of different emission sectors in the original inventory. For emission calculations, means instead of geometric means of EFs and $R_{C/S}$ were used. Monte Carlo simulations were used to characterize the uncertainty in the compiled inventories caused by the activity and EF data. Since the calculated uncertainties are quantified for individual source types for each grid cell in a given month, they represent the true uncertainty and also characterize the temporal and spatial variability. 10,000 simulations were performed using randomly plotted activity and EF data from various sources. Parameters were normally or log-normally distributed, with known (based on data reported in the literature) or assumed coefficients of variation (10% for residential biomass fuels and 5% for all other sources).

To ensure the accuracy of the experimental data, strict QA/QC was performed during sample collection and analysis processes, including simultaneous collection and analysis of field blanks and 10–20% duplicate samples.

## Data availability

All data generated in this study are available in the Supplementary Information and have been deposited in the Zenodo database available at https://www.zenodo.org/record/8340969.

## Code availability

The dominance analysis package used in this study is developed based on the R programming environment. Basic codes and usage instructions can be found at https://www.rdocumentation.org/packages/dominanceanalysis/versions/2.0.0. The specific code with sample data have been deposited in the Zenodo database available at https://www.zenodo.org/record/8340969.

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

## Acknowledgements

This study was funded by the National Natural Science Foundations of China (grant NOs. 42192514, 91744203, 42177086, and 42030715) and Guangdong Basic and Applied Basic Research Foundation (2017BT01Z134).

## Author contributions

Conceptualization: Ju.C., H.J., Y.C., and G.Z.; Sample collection: Ju.C. and Z.L.; Performed experiments: Ju.C., H.J., and Z.L.; Methodology: Ju.C., H.J., Y.H., Y.C., H.S., J.S., J.L., Y.Z., Ji.C., and G.Z.; Writing—review & editing: Ju.C., H.J., and Y.C.; Supervision: Y.C., H.S., R.W., J.S., J.L., Y.Z., and G.Z.; Funding acquisition: Y.C. and G.Z.

## Competing interests

The authors declare no competing interests.
