## [Peer Review File · Nature Communications]

Char Dominates Black Carbon Aerosol Emission and Its Historic Reduction in ChinaReviewer #1 (Remarks to the Author):

Comments on "Char Dominates Black Carbon Aerosol Emission and Its Historic Reduction in China" by Cai et al.

Char and soot, the two subtypes of black carbon (BC), have different physicochemical properties and effects, thus dividing BC emission inventory into char and soot inventories separately is important to better understand the driving forces behind BC emission reductions. This manuscript conducted extensive field and laboratory experiments to obtain a comprehensive multi-sector database of emission factors of char and soot for constructing their emission inventories. The updated emission factors along with the reconstructed historical (1960-2017) emission inventories in China, contribute to our understanding of BC emissions. The findings highlight the dominance of char in BC emissions and its substantial historical decrease, which were attributed to changes in energy structure, combustion technology, and emission standards. Overall, the manuscript offers a valuable contribution to assessing the effects of BC on climate, environment, and human health, and provides insights for future mitigation strategies.

Specific comments:

1. There are too many abbreviations throughout the manuscript, making the text not easy to read and comprehend. While abbreviations can be useful for efficiency and brevity, it is important to strike a balance. In addition, full names of some abbreviations (e.g., TIS, IIS, HBS in the supplement) are not given in the text.

2. Lines 62-64, please add the reference of Han et al., 2010, ACP. This reference is the first one that reported the char/soot ratios (in the name of char and soot, rather than from EC fractions) in different sources.

3. For the emission sector of industry, detailed information about classification should be given in the manuscript. For example, did pharmaceutical factories in Text S1 correspond to the pharmaceutical plants in Table S1? Why wasn't it included in the ICB or PCB?

4. Line 133-134: check the sentence, "when dividing BC emission inventory into char and soot subgroups" instead of "when calculating the dividing BC emission inventory into char and soot subgroups".

5. The ratio of annual char to soot emissions (RCE/RSE) is calculated by dividing the annual char emissions to soot emissions. Thus, uncertainty of char and soot emissions (present in Fig.2) is propagated to the RCE/RSE. Besides the best estimate of RCE/RSE shown in Fig. 2, uncertainties of RCE/RSE are also important to understand the variations of RCE/RSE. The reviewer suggests the authors to add the uncertainties of RCE/RSE in Fig.2a.

6. In Figs. 2c and 3d, it is not clear to me, what does each emission source include? For example, emissions sources "Other anthropogenic" and "Energy sector" in the figure legend are not mentioned and explained in the text; some emission sectors, e.g., off-road vehicles, are discussed in the text, but not shown in the Figs. 2c and 2d; From discussions in Line160-170, beehive coke oven is included in the industrial emission sector, but it is not clear in Fig.2. Taking the above concerns into account, the reviewer suggests the authors to explain each emission source, at least in the figure caption.

7. Line 274. "Implication: future BC reduction should pay more attention on char emission." I would like to recommend this be limited to the topic of air pollution control. In the field of climate change, soot exhibits stronger light absorption capacity than char and with little spectral dependence, which suggest that in climate topic much more attentions should also be paid to soot.

8. Line 293-294 "However, advanced stoves have less impact on BC and char reduction compared to traditional ones (Fig. S2)." Here, the Figure citation is incorrect. Fig. S2 shows the results of char and soot emissions in China between atmospheric and sediment cores, while having nothing to do with different types of stoves.

9. In Figure S2, the data sources are missing for sediment cores.

10. Method section: Dilution ratios in this study were 10–20. How was the dilution ratio considered and selected?

11. Line 350: Here, better to explain that EC is the mass-based analog of BC, to link the EC fractions, i.e., EC1-PyC and EC2+EC3 to char-BC and soot-BC, respectively.

12. Line 363–368: the "Dominance Analysis" in R, could the authors explain this method in more details? For example, what's the principle of this analysis? what's the input data?

Reviewer #2 (Remarks to the Author):

This study examined more than 400 black carbon (BC) emission events from various combustion sources, including residential, mobile and industrial sources. It determined the emission factors and ratio characteristics of char and soot for different emission sources. By analyzing the char/soot ratios, a historical BC emission inventory for China since 1960 was reconstructed. The results showed that char accounted for the majority of annual BC emissions and reductions in China. The historical reduction of char emissions can be attributed to the rapid changes in energy structure, combustion technology and emission standards in recent decades. Based on the inter-annual variations of China's BC emissions and char/soot ratios, the history was divided into three major stages, which can serve as valuable references for other countries at different stages of development. In addition to the emission inventory, the authors suggested that further focus of BC emission reduction should be placed on char, as it is mainly produced by lower-temperature combustion and is relatively easy to reduce compared to soot.

Given the general uncertainty in BC estimates at regional and global scales often attributed to ageing and coating of BC particles in previous studies, this research brings a fresh perspective by focusing on the variability of BC itself. The approach of the article is innovative and the results and conclusions presented are of significant importance. The work is original, supported by the authors' own BC measurements, and other data sources are appropriately referenced and available online. No inconsistencies were found in the manuscript or supplementary materials, and the methods used are robust and well explained. Given its scientific merit, I see no impediment to the publication of this manuscript.

I recommend to address the following comments prior to publication in Nature Communications.

1) Please ensure that abbreviations used in the text are re-presented in the image to ensure the self-explanatory nature of the pictures for readers to understand their meaning.

2) In terms of English writing and word usage, this article still has some minor issues. I will list them here one by one, hoping the author can further proofread their text.

Lines 29, 51: "On global scale" should be "at global scale".

Lines 33, 36, 203: Some nouns should be plural.

Lines 41, 45, 55, 104, 149: Some proper nouns should be prefixed with "the".

Lines 63-64: Please check the font issue in the reference citations.

Line 69: "Mountain area" should be "mountainous areas"

Lines 101, 111, 264: Incorrect use of prepositions and passive voice

Line 158: Avoid semantic repetition and excessive use of abbreviations.

Line 197: Add a space after the full stop.

Line 208: Problems in describing directions within China; eastern, southeastern, etc. would be more appropriate.

3) The author in the text uses RC/S and RCE/CE to represent the ratio of char and soot in emission sources and annual emissions. However, for environmental samples, directly using RCE/SE may cause misunderstandings, so some additional explanations are needed (Lines 281- 283).

Reviewer #3 (Remarks to the Author):

Black carbon has significantly important impacts on air quality, climate change and human health. As one widely concerned SLCP with health co-impacts, it is imperatively valuable and important to scientifically assess fates and impacts of BC in environment. Inventory of BC is the basic task with a high priority in current scientific researches and evaluation of BC. BC is initially defined from its absorption light property in atmospheric science, while studies in other areas like geochemistry and soil sciences have distinct definitions with different terms such as soot, carbon black, elemental carbon (EC), etc., but now there are growing discussions on the chemical properties of BC in atmospheric science. Char- and soot-EC (or BC) is therefore defined and studied in a few studies. Different physiochemical properties and consequent climate and health impacts are expected.

This study, for the first time, developed char- and soot-BC inventories in China by separating these two parts from the BC inventory with experiment-based ratios of Char/soot, and further found that char dominates BC emissions in the country and contributed largely to the reduced BC emissions. Results of the study are important for source controls and policy making on the control of biomass and fossil fuel combustions in different sectors. I think the study is valuable and important with notable scientific significance. The manuscript is also clearly presented and well organized. In prior to its acceptance for publication, I'd like to suggest the authors considering the following issues in revision:

1. Char- and soot-BC are operationally defined, that is the values and ratios of these two are closely associated with the analysis method (using IMPROVE protocol and TOR method in the present study). This would mean different values may be obtained when other analysis methods were adopted, which directly affect the quantitative results of the study. It is necessary to assess and discuss uncertainties and potential biases associated with the analytical methods and the definitions of these terms.
2. The study measured and reported EFs from different sources (main types of BC emissions), but in developing inventories, ratios of char/soot were incorporated into a previously published inventory. Why not using directly measured EFs? By comparing EFs in the present study with literature studies, obvious differences were observed (R&D line and 100-105 and figure s1), and the authors explained this by different combustion and control technologies. Using the ratio in developing char- and soot-BC inventories is from the assumption of combustion processes and control technologies did not affect or change the ratio of char/soot, which may be not valid in most cases. This part needs to be clearly evaluated and probably updated.
3. Emission factors are from laboratory and field-based measurements. It has been well recognized that results from these two approaches would be rather different, depending on methods and varying among fuel types and combustion facilities in different sectors. The authors are suggested to comment on the differences on EFs from these two approaches, and how to integrate data from different methods for different sources, as well as its contribution to the inventory uncertainty.
4. The study suggested that future reductions should focus on char, in China and the world. Two main concerns here: with significant reductions of char-BC in China, why it would be still the main focus in future BC reduction but not the remaining part? Secondly, without global inventories including its historical trends, it is too simplistic to conclude that the world BC reductions should focus on char based on the present analysis.
5. Are these direct evidences on the distinct light properties of char and soot-BC in literature studies? some references cited in the introduction are not from the area of atmospheric science, in another words, they had different definition and quantitation methods of char and soot in comparison with the present method.

Some specific ones:

Abstract – this part is suggested to be written. For example, the first two sentences could be shortened leaving more spaces for key results and findings. "a significant dataset"?

Line 52-55, as mentioned, are direct studies on different physiochemical properties and especially light properties of char and soot-BC that is defined and analyzed by the same method as you did here? In other areas, "soot" or "char" being separated from combustion exhausts are rather different from those from the operational definition here.

Line 58, "a single rule" or a single ratio? The rule is nearly the same and straightforward that is the ratio of char/soot for different sources, by fuels and combustion technologies, and the important issue here is specific and reliable (accurate) values of those ratios for different sources.

Line 65, which year? What about emissions from India, as the Chinese emissions are reducing?

Line 80, in these 400 events (including repeats?), it is more necessary to clearly show the sample sizes for each sector, and those from laboratory and field tests, and the representativeness of these events.

Line 105-108, the authors are suggested to carefully exam these data and well explain the differences observed. In figure s1, EFs from briquette coals were higher, and EF differences in brick kiln and coke oven were too large, in nearly 3-4 orders of magnitude. The authors stated many studies in past were prior to 2012, but in fact as seen in Table s2, there are few studies prior to 2012. The explanation of fuel composition and different after-treatment technology are too general and cannot well explain the difference.

Line 110-120, please add uncertainty range of these numbers.

Line 123, how to ensure results are "more accurate"?

Line 141-142, rethink this assumption

Line 150 -170, are BC emissions reported here are from the present calculation, or the paper of Xu et al.?

Line 173, may be helpful to add the reduction percentages, and variations in these estimates

Line 176-180, it is not very clear in the logistic relationship between the study cited here and the discussion in this part. The value was 1.41 (despite of different analytical methods and term definition) in 2019, that was close to the 1.37 in the 2001 in your inventory

Line 201-212, since difference methods and even definition of terms here, how to be sure these results, evening the trends, can be compared and mutually corroborated? It is suggested to comment on the uncertainty and cautions in exploring the results here.

Line 229-273, suggest to shorten and be more specific as some were discussed in previous publications resulting in limited new information here (and some still needs more experimental and modelling evidences to valid the mechanisms).

Line 310, if solid fuels were not burned in stoves, which energy would be used in the residential sector? Will that produce relatively more soot-BC if modern energies like gas fuels and electricity from coal-fired plants were used?

Line 310-315, do you mean also burn solid fuels in boilers in India and enforcing vehicle emissions in US? BC source, as well as EFs in other countries could be very different from that in China, and vehicle emissions were affected by not only fuel types but road conditions that are very different between China and US. How about the accuracy in the estimated numbers like 64%, 75%, and 53%? The calculation method and future reduction scenarios of BC emission from India and US is not available in the Method section.

Line 372, what do you mean "timeliness" here?

Line 372-376, in your present study or Xu et al.,?

Line 379 and 381, which parameters? Please be more specific

Figure s1, what are the triangles in the figure? What about variations in these EFs that are usually large? is 'Historical EF' the "previous inventory" in figure caption? "measured data"- "updated inventory"? inventory or only EFs?

Response to reviewers

Reviewer #1:

[General comment] Char and soot, the two subtypes of black carbon (BC), have different physicochemical properties and effects, thus dividing BC emission inventory into char and soot inventories separately is important to better understand the driving forces behind BC emission reductions. This manuscript conducted extensive field and laboratory experiments to obtain a comprehensive multi-sector database of emission factors of char and soot for constructing their emission inventories. The updated emission factors along with the reconstructed historical (1960-2017) emission inventories in China, contribute to our understanding of BC emissions. The findings highlight the dominance of char in BC emissions and its substantial historical decrease, which were attributed to changes in energy structure, combustion technology, and emission standards. Overall, the manuscript offers a valuable contribution to assessing the effects of BC on climate, environment, and human health, and provides insights for future mitigation strategies.

[General response] Thank you for your positive evaluation of our manuscript and the acknowledgment of its scientific significance. Your feedback and suggestions will help improve our work. We have made revisions following the comments (corrections are marked in the revised manuscript), and the responses are shown below, with responses are in Blue and revised parts are in Red.

[Specific comment]

[Comment 1] There are too many abbreviations throughout the manuscript, making the text not easy to read and comprehend. While abbreviations can be useful for efficiency and brevity, it is important to strike a balance. In addition, full names of some abbreviations (e.g., TIS, IIS, HBS in the supplement) are not given in the text.

[Response] Thank you very much for providing your professional advice. We have removed some acronyms that were used only once (appears primarily in the discussion and supplement section) and ensured that each acronym has its corresponding full name. Here are the specific revised versions.

Lines 256-260 in the revised Manuscript:

We found that the high time-resolved $R_{C/S}$ were closely proportional to higher ratios of high-molecular-weight (4-6 ring) to low-molecular-weight (2-3 ring) PAHs as well as higher ratios of refractory organic carbon to volatile organic carbon of OC fractions.

Lines 57-60 in the revised Supporting Information:

Biomass is burned uniformly in brick stoves, while lump coal and honeycomb coal are burned in separate stoves (traditional iron stove (TIS), improved iron stove (IIS) and honeycomb briquette stove (HBS)). Brick stoves is about 50 cm high and equipped with 1-3 iron pots.

Lines 79-83 in the revised Supporting Information:

At the same time, particulate matter was collected from the exhaust of 6 forklift trucks of different tonnages and aftertreatment technology levels (with or without diesel particulate filter (DPF)) under different driving conditions (idling, unloaded and loaded)

during January-December 2021 using the same sampling method, detailed in Table S8.

Lines 94-95 in the revised Supporting Information:

A total of 76 particle samples were collected from three ocean-going vessels (OGV, an Aframax oil tanker, a Newcastlemax bulk carrier and a Capesize bulk carrier, detailed parameters in Table S9) between December 2020 and August 2021.

Lines 126-129 in the revised Supporting Information:

The diluted flue gas was subsequently introduced to a filter sampler (quartz fiber filter; $d = 90$ mm) to collect the total suspended particles. Each combustion experiment was repeated twice for all fuel/temperature combinations to collect samples.

Lines 131-132 in the revised Supporting Information:

Text S2. Calculation of emission factors (EFs) and modified combustion efficiency (MCE)

Lines 171-172 in the revised Supporting Information:

For OGVs, both the generator and the main engine are kept on throughout the journey, and the usage of heavy fuel oil (HFO) is about 4 times that of marine gas oil (MGO).

[Comment 2] Lines 62-64, please add the reference of Han et al., 2010, ACP. This reference is the first one that reported the char/soot ratios (in the name of char and soot, rather than from EC fractions) in different sources.

[Response] Thank you very much for your suggestion. It was precisely this literature that inspired our initial ideas and creativity. Here is the revised content:

Lines 56-59 in the revised Manuscript:

However, separating BC into char and soot subgroups needs specific and reliable values of char to soot ratios for different emission sectors, resulting from differences in fuel composition, combustion, and after-treatment technologies¹

- 1 Han, Y. M., Cao, J. J., Lee, S. C., Ho, K. F. & An, Z. S. Different characteristics of char and soot in the atmosphere and their ratio as an indicator for source identification in Xi'an, China. *Atmos. Chem. Phys.* **10**, 595-607, doi:10.5194/acp-10-595-2010 (2010).

[Comment 3] For the emission sector of industry, detailed information about classification should be given in the manuscript. For example, did pharmaceutical factories in Text S1 correspond to the pharmaceutical plants in Table S1? Why wasn't it included in the ICB or PCB?

[Response] Thank you for the valuable suggestions. We have made revisions to the content in the Supporting Information regarding sampling from industrial sources and power plants in Text S1, removing unclear and repetitive sections. In addition, we have provided additional details on sample characteristics from different emission sources in the introduction section of the manuscript and add Table 1 of our sample collection details.

Lines 100-108 in the revised Supporting Information:

The real-world measurement of ICB and PCB were conducted in Hebi City, Henan

Province, from 1 to 5 November 2018. Hebi City is one of the major air pollution transmission channels in the Beijing-Tianjin-Hebei region ("2+26" cities). In summary, for ICB, we collected emission samples from a coal-fired boiler, two coking plants, and two pharmaceutical factories in the vicinity of Hebi City, based on the local industrial source emission inventory provided by the Hebi Environmental Protection Bureau. As for power plant sources, we selected two representative local biomass power plants and a coal-fired power plant for sampling. The samplers were installed at the flue gas outlet after the dust removal.

Lines 82-87 in the revised Manuscript:

In this study, we collected combustion event data from three different sources, including 122 samples from RS, 160 samples from VE, and 12 samples from CB (specific information shown in Table 1). Based on this data, we established a comprehensive multi-sector database of char and soot emission factors. By further combining 105 sets of results from laboratory stove simulations, we identified key factors influencing the variation in char and soot emissions and explained the differences in their formation pathways.

[Comment 4] Lines 133–134: check the sentence, “when dividing BC emission inventory into char and soot subgroups” instead of “when calculating the dividing BC emission inventory into char and soot subgroups”.

[Response] Thank you for carefully reading our manuscript. We have fixed the language in the manuscript. The specific amendments are as follows:

Lines 139-141 in the revised Manuscript:

It also suggests that the actual proportion of char and soot in EF_{BC} for each source should be taken into account when calculating the dividing BC emission inventory into char and soot subgroups

[Comment 5] The ratio of annual char to soot emissions ($R_{CE/SE}$) is calculated by dividing the annual char emissions to soot emissions. Thus, uncertainty of char and soot emissions (present in Fig. 2) is propagated to the $R_{CE/SE}$. Besides the best estimate of $R_{CE/SE}$ shown in Fig. 2, uncertainties of $R_{CE/SE}$ are also important to understand the variations of $R_{CE/SE}$. The reviewer suggests the authors to add the uncertainties of $R_{CE/SE}$ in Fig.2a.

[Response] Thank you for your important suggestions to improve the visibility of our figures. As you pointed out, in addition to the best estimate, the uncertainty of $R_{CE/SE}$ is also an important parameter in understanding the historical emissions inventory of char and soot. Following your suggestion, we have redrawn Fig. 2a and added annotations to the legend section. The redrawn Fig. 2a is as follow:

[Comment 6] In Figs. 2c and 3d, it is not clear to me, what does each emission source include? For example, emissions sources “Other anthropogenic” and “Energy sector” in the figure legend are not mentioned and explained in the text; some emission sectors, e.g., off-road vehicles, are discussed in the text, but not shown in the Figs. 2c and 2d; From discussions in Lines 160–170, beehive coke oven is included in the industrial emission sector, but it is not clear in Fig.2. Taking the above concerns into account, the reviewer suggests the authors to explain each emission source, at least in the figure caption.

[Response] Thank you for your important suggestions to improve the visibility of our figures. We have updated the legend of Figure 2. The specific modifications are as follows.

Fig. 2 Total BC and char to soot ratio of anthropogenic source emissions in China from 1960-2017 (a), temporal trends of annual emission of char and soot (b) (Shaded areas are uncertainty intervals of Char and Soot, represented by quartiles). The percentage contribution of each emission source to char (c) and soot (d) (" Other anthropogenic " refers to emissions from non-road transportation, "Energy sector" refers to emissions from power plants, specifically PCBs mentioned in the text.)

[**Comment 7**] Line 274. "Implication: future BC reduction should pay more attention on char emission." I would like to recommend this be limited to the topic of air pollution control. In the field of climate change, soot exhibits stronger light absorption capacity

than char and with little spectral dependence, which suggest that in climate topic much more attentions should also be paid to soot.

[Response] Thank you very much for your valuable suggestions. Based on your concerns, we would like to elaborate on two points. Firstly, in this section, our emphasis is not solely on highlighting the importance of char in the field of air pollution or climate change, but rather on emphasizing targeted combustion technology improvements for different emission sources. With the advancement of combustion technology, char primarily originates from low-temperature combustion emissions, making it a significant part of BC reduction and an area of focus for future emissions reduction efforts. Secondly, although the improvement in combustion technology may lead to an increase in the proportion of soot in BC, both the emission factor and the total emissions of Soot will also decrease significantly. Therefore, despite soot potentially having a stronger light-absorbing effect compared to char, the overall light-absorbing capacity of the atmosphere will still decrease noticeably.

[Comment 8] Lines 293–294 “However, advanced stoves have less impact on BC and char reduction compared to traditional ones (Fig. S2).” Here, the Figure citation is incorrect. Fig. S2 shows the results of char and soot emissions in China between atmospheric and sediment cores, while having nothing to do with different types of stoves.

[Response] We appreciate your careful reading of our article. In the Supporting Information section, we omitted the images depicting changes in char and soot emission

factors of simulated domestic stoves under different combustion temperatures and oxygen supply conditions. This has caused confusion for you and other readers during the reading process. We have now included the missing image in the revised Supporting information section, along with corresponding annotations (Fig. S3). The specific amendments are as follows:

Figure S3 EF_{BC} and R_{C/S} of modified RSFC under different chamber conditions

[Comment 9] In Figure S2, the data sources are missing for sediment cores.

[Response] Thank you for your professional suggestion. We made corresponding modifications to Figure S2, and the specific changes are as follows.

Figure S2. Comparison of char (a), soot (b) emission and char / soot (c) in China between atmospheric and sediment cores (South yellow sea: SYS, North yellow sea: NYS, Atmospheric annual emission: ATMOs), detailed information of the sediment cores could be found in Fang et al².

2 Fang, Y. *et al.* Spatiotemporal Trends of Elemental Carbon and Char/Soot Ratios in Five Sediment Cores from Eastern China Marginal Seas: Indicators of Anthropogenic Activities and Transport Patterns. *Environmental Science & Technology* **52**, 9704-9712, doi:10.1021/acs.est.8b00033 (2018).

[Comment 10] Method section: Dilution ratios in this study were 10–20. How was the dilution ratio considered and selected?

[Response] Thank you for your professional advice. We have added additional discussion in the Methods section to provide a more detailed explanation of this issue.

Lines 337-345 in the revised manuscript:

When selecting the dilution ratio, we took into account both the water vapor concentration and the sample concentration. Firstly, for engine exhaust from vehicles as well as industrial emissions, the water vapor concentration in the smoke is close to 100%, and water vapor condenses at a concentration of around 10%. Therefore, we chose the lowest dilution ratio of 10 to ensure that water vapor does not condense in the smoke pathway and to ensure that the collected sample quantity does not fall below the instrument's detection limit. Secondly, for household stoves with high particulate matter concentrations in the smoke, we chose a relatively higher dilution ratio of 20 to ensure that the sample does not exceed the loading capacity of the filter membrane.

[Comment 11] Line 350: Here, better to explain that EC is the mass-based analog of BC, to link the EC fractions, i.e., EC1-PyC and EC2+EC3 to char-BC and soot-BC, respectively.

[Response] Your feedback is crucial for improving the readability of our manuscript. We have provided additional clarification on the definition of black carbon in the

methodology section. The specific amendments are as follows.

Line 351 in the revised Manuscript:

Black carbon (quantified as BC using optical methods and as EC using thermal optical methods, these two concepts can be used interchangeably)

[Comment 12] Lines 363–368: the “Dominance Analysis” in R, could the authors explain this method in more details? For example, what’s the principle of this analysis? what’s the input data?

[Response] Your feedback is very helpful for improving the readability of our article. In the methodology section, we have provided supplementary information on the background and principles of the advantage analysis method to give readers a basic understanding of this approach. Here is the revised version.

Lines 375-383 in the revised Manuscript:

In scientific research, an important issue is to determine the specific contribution of different explanatory variables to the variance of the dependent variable. In 2007, Israeli proposed a method called importance analysis based on previous studies. This method quantitatively determines the contribution of different explanatory variables to the determination coefficient (r^2) and variance of the dependent variable in linear regression, by inputting multiple explanatory variables and the dependent variable³. In recent years, this method has also been widely used in the environmental field to determine the potential impacts of various influencing factors on emissions⁴⁻⁶.

- 3 Israli, O. A Shapley-based decomposition of the R-square of a linear regression. *The Journal of Economic Inequality* **5(2)**, 199-212 (2007).
- 4 Zhang, L. *et al.* Mass Absorption Efficiency of Black Carbon from Residential Solid Fuel Combustion and Its Association with Carbonaceous Fractions. *Environmental Science & Technology* **55**, 10662-10671, doi:10.1021/acs.est.1c02689 (2021).
- 5 Zhang, L. *et al.* Optically Measured Black and Particulate Brown Carbon Emission Factors from Real-World Residential Combustion Predominantly Affected by Fuel Differences. *Environmental Science & Technology* **55**, 169-178, doi:10.1021/acs.est.0c04784 (2021).
- 6 Liu, Z. *et al.* Emission Characteristics and Formation Pathways of Intermediate Volatile Organic Compounds from Ocean-Going Vessels: Comparison of Engine Conditions and Fuel Types. *Environmental Science & Technology* **56**, 12917-12925, doi:10.1021/acs.est.2c03589 (2022).

Reviewer #2:

[General comment] This study examined more than 400 black carbon (BC) emission events from various combustion sources, including residential, mobile and industrial sources. It determined the emission factors and ratio characteristics of char and soot for different emission sources. By analyzing the char/soot ratios, a historical BC emission inventory for China since 1960 was reconstructed. The results showed that char accounted for the majority of annual BC emissions and reductions in China. The historical reduction of char emissions can be attributed to the rapid changes in energy structure, combustion technology and emission standards in recent decades. Based on the inter-annual variations of China's BC emissions and char/soot ratios, the history was divided into three major stages, which can serve as valuable references for other countries at different stages of development. In addition to the emission inventory, the authors suggested that further focus of BC emission reduction should be placed on char, as it is mainly produced by lower-temperature combustion and is relatively easy to reduce compared to soot.

Given the general uncertainty in BC estimates at regional and global scales often attributed to ageing and coating of BC particles in previous studies, this research brings a fresh perspective by focusing on the variability of BC itself. The approach of the article is innovative and the results and conclusions presented are of significant importance. The work is original, supported by the authors' own BC measurements, and other data sources are appropriately referenced and available online. No inconsistencies were found in the manuscript or supplementary materials, and the methods used are

robust and well explained. Given its scientific merit, I see no impediment to the publication of this manuscript.

I recommend to address the following comments prior to publication in Nature Communications.

[General response] Thank you for thoroughly reviewing our work and acknowledging the scientific and practical value it holds. Your positive assessment of our work will encourage us to further explore this direction in-depth. We have made revisions following the comments (corrections are marked in the revised manuscript), and the responses are shown below, with responses are in Blue and revised parts are in Red.

[Comment 1] Please ensure that abbreviations used in the text are re-presented in the image to ensure the self-explanatory nature of the pictures for readers to understand their meaning.

[Response] Thank you very much for pointing out the important issues in our manuscript. We have revised the captions for each image to ensure that all abbreviations used in the figure legends are fully explained. Additionally, we have conducted a second check to ensure that the quality and content of each image meet the standards of Nature Communications.

[Comment 2] In terms of English writing and word usage, this article still has some minor issues. I will list them here one by one, hoping the author can further proofread their text.

[Response] Thank you once again for your meticulous reading of our article. We will be happy to edit the text further, based on helpful comments from the reviewers. The modified text content is as follows:

Lines 29, 51: "On global scale" should be "at global scale".

Lines 48-50 in the revised Manuscript:

To accurately estimate the impact of BC, emission factors of BC for various sectors have been measured and bottom-up emission inventories have been developed at national and global scales in recent decades.

Lines 33, 36, 203: Some nouns should be plural.

Lines 30-31 in the revised Manuscript:

Here, we present a substantial dataset of char and soot emission factors derived from field and laboratory measurements.

Lines 197-198 in the revised Manuscript:

In general, it can be concluded that char dominates the annual BC emissions and its huge historical reduction in China.

Lines 41, 45, 55, 104, 149: Some proper nouns should be prefixed with "the".

Lines 40-41 in the revised Manuscript:

Black carbon (BC) aerosol, derived from the incomplete combustion of biomass and fossil fuels

Lines 43-44 in the revised Manuscript:

The "dome effect" of BC lowers the development of the planetary boundary layers, leading to higher instances of extreme haze pollution.

Lines 53-56 in the revised Manuscript:

Although they are originally separated by different temperatures in the Interagency Monitoring of Protected Visual Environments (IMPROVE) protocol and the thermal-optical reflectance (TOR) method for organic and elemental carbon analysis

Lines 104-107 in the revised Manuscript:

This variation of EF_{BC} among emission sectors could be jointly explained by the fuel composition (e.g., the difference between solid fuels and petroleum products), combustion technology (e.g., coal combustion in stoves versus boilers) and emission standards (e.g., the difference between NVE and OVE).

Lines 151-153 in the revised Manuscript:

an important industrial source that existed mainly in 1990-2000 with combustion technologies and after-treatment measures much closer to RCC than the current ICB.

Lines 63-64: Please check the font issue in the reference citations.

Lines 63-64 in the revised Manuscript:

With the rapid development of the economy, China became the world's largest emitter of BC in 2000, accounting for about one-fourth of the global BC emissions.

Line 69: "Mountain area" should be "mountainous areas"

Lines 69-71 in the revised Manuscript:

including residential stove combustion (RS) of solid fuels such as biomass and coal, which occurs in rural or undeveloped mountainous areas for heating and cooking,

Lines 101, 111, 264: Incorrect use of prepositions and passive voice

Lines 102-104 in the revised Manuscript:

while the CB sector (ICB at 0.86 ± 0.82 mg/kg and PCB at 0.11 ± 0.12 mg/kg) was lower by three orders of magnitude (Fig. 1a).

Lines 116-117 in the revised Manuscript:

Therefore, EF_{BC} data used for emission inventory calculation needs to be updated in a timely manner.

Lines 256-257 in the revised Manuscript:

Secondly, different by-products or intermediate products are generated between char and soot formation pathways.

Line 158: Avoid semantic repetition and excessive use of abbreviations.

We have removed some acronyms that were used only once (appears primarily in the discussion and supplement section) and ensured that each acronym has its corresponding full name. Here are the specific revised versions.

Lines 256-260 in the revised Manuscript:

We found that the high time-resolved $R_{C/S}$ were closely proportional to higher

ratios of high-molecular-weight (4-6 ring) to low-molecular-weight (2-3 ring) PAHs as well as higher ratios of refractory organic carbon to volatile organic carbon of OC fractions.

Lines 57-60 in the revised Supporting Information:

Biomass is burned uniformly in brick stoves, while lump coal and honeycomb coal are burned in separate stoves (traditional iron stove (TIS), improved iron stove (IIS) and honeycomb briquette stove (HBS)). Brick stoves is about 50 cm high and equipped with 1-3 iron pots.

Lines 79-83 in the revised Supporting Information:

At the same time, particulate matter was collected from the exhaust of 6 forklift trucks of different tonnages and aftertreatment technology levels (with or without diesel particulate filter (DPF)) under different driving conditions (idling, unloaded and loaded) during January-December 2021 using the same sampling method, detailed in Table S8.

Lines 94-95 in the revised Supporting Information:

A total of 76 particle samples were collected from three ocean-going vessels (OGV, an Aframax oil tanker, a Newcastlemax bulk carrier and a Capesize bulk carrier, detailed parameters in Table S9) between December 2020 and August 2021.

Lines 126-129 in the revised Supporting Information:

The diluted flue gas was subsequently introduced to a filter sampler (quartz fiber filter; $d = 90$ mm) to collect the total suspended particles. Each combustion experiment was repeated twice for all fuel/temperature combinations to collect samples.

Lines 131-132 in the revised Supporting Information:

Text S2. Calculation of emission factors (EFs) and modified combustion efficiency (MCE)

Lines 171-172 in the revised Supporting Information:

For OGVs, both the generator and the main engine are kept on throughout the journey, and the usage of heavy fuel oil (HFO) is about 4 times that of marine gas oil (MGO).

Line 197: Add a space after the full stop

Lines 207-209 in the revised Manuscript:

In the BC emission reduction of 309 Gg during 2001-2017, char emission decreased by 265 Gg and contributed 86% of the BC reduction, while soot decreased by 44 Gg and contributed only 14% of BC reduction in this period.

Line 208: Problems in describing directions within China; eastern, southeastern, etc. would be more appropriate.

Lines 207-209 in the revised Manuscript:

These cores, including Taihu Lake and Chaohu Lake in Eastern China, Nam-Co and Dagzo-Co Lake in the Tibetan area, Dianchi Lake in Southeastern China, and the East China Margin Sea, showed a continuous increase in BC and char concentrations starting from the 1960s, peaking around 1995, and then declining significantly.

[Comment 3] The author in the text uses $R_{C/S}$ and $R_{CE/SE}$ to represent the ratio of char

and soot in emission sources and annual emissions. However, for environmental samples, directly using $R_{CE/SE}$ may cause misunderstandings, so some additional explanations are needed (Lines 281- 283).

[Response] Thank you for providing valuable feedback. In the manuscript, we used $R_{C/S}$ to represent the ratio of char to soot directly measured, while $R_{CE/SE}$ was used to denote data that underwent secondary calculations. We made the following specific modifications:

Line 282-284 in the revised Manuscript:

In these countries, significantly high $R_{C/S}$ values were often observed from urban ambient $PM_{2.5}$ samples (e.g., $R_{C/S}$ exceeded 2.0 in urban environmental samples in India).

Reviewer #3

[General comment] Black carbon has significantly important impacts on air quality, climate change and human health. As one widely concerned SLCP with health co-impacts, it is imperatively valuable and important to scientifically assess fates and impacts of BC in environment. Inventory of BC is the basic task with a high priority in current scientific researches and evaluation of BC. BC is initially defined from its absorption light property in atmospheric science, while studies in other areas like geochemistry and soil sciences have distinct definitions with different terms such as soot, carbon black, elemental carbon (EC), etc., but now there are growing discussions on the chemical properties of BC in atmospheric science. Char- and soot-EC (or BC) is therefore defined and studied in a few studies. Different physiochemical properties and consequent climate and health impacts are expected.

This study, for the first time, developed char- and soot-BC inventories in China by separating these two parts from the BC inventory with experiment-based ratios of Char/soot, and further found that char dominates BC emissions in the country and contributed largely to the reduced BC emissions. Results of the study are important for source controls and policy making on the control of biomass and fossil fuel combustions in different sectors. I think the study is valuable and important with notable scientific significance. The manuscript is also clearly presented and well organized. In prior to its acceptance for publication, I'd like to suggest the authors considering the following issues in revision:

[General response] Thank you for thoroughly reviewing our work and acknowledging

the scientific and practical values it holds. Your positive assessment of our work will encourage us to further explore this direction in-depth. We have made revisions following the comments (corrections are marked in the revised manuscript), and the responses are shown below, with responses are in Blue and revised parts are in Red.

[General Comment 1] Char- and soot-BC are operationally defined, that is the values and ratios of these two are closely associated with the analysis method (using IMPROVE protocol and TOR method in the present study). This would mean different values may be obtained when other analysis methods were adopted, which directly affect the quantitative results of the study. It is necessary to assess and discuss uncertainties and potential biases associated with the analytical methods and the definitions of these terms.

[Response] Thanks for your professional question. The question you raised is crucial for a deeper understanding of the methodological definition differences between char and soot. Due to space constraints, we cannot provide a detailed answer in the manuscript, but we will provide you with a comprehensive response here.

In previous studies, char and soot were actually considered as part of the continuum of combustion products in soil science definitions⁷. Due to morphological differences, they can be easily distinguished qualitatively but are difficult to analyze quantitatively, which prompted scientists to search for new quantitative methods⁸. In 2007, Han et al. first quantified char and soot using the IMPROVE protocol and TOR method⁹. They used pine wood char materials prepared at different temperatures, *n*-

hexane soot generated from acetylene flame, and a pure Soot reference material RS683 produced by Cabot Corporation as standard samples for char and soot. The char sample in the standard samples appeared as the EC1-POC portion in the IMPROVE protocol, soot generated from the acetylene flame appeared as the majority of EC2 and a minority of EC3, while the Soot reference sample showed a broad EC2 peak and a sharp EC3 peak. Based on the results of these standard samples, the IMPROVE protocol under TOR method has become widely recognized as the standard for quantitative analysis of char and soot for over a decade. It is worth mentioning that this standard using IMPROVE protocol and TOR method on OC/EC analyzers is now not only applied to atmospheric samples but also to BC samples collected in other domains such as the hydrosphere and pedosphere, and is already widely used in academia^{2,10-12}. More importantly, based on a series of studies on the nature of BC, the results indicate that under the IMPROVE protocol, char and soot are not just operational definitions, but they also exhibit significant differences in their physicochemical properties^{2,13}. Therefore, at present, the IMPROVE protocol is the only method available to classify black carbon into Char and Soot components. Other analysis methods applied on OC/EC analyzers, such as EUSAAR, cannot be used for char and soot differentiation.

- 2 Fang, Y. *et al.* Spatiotemporal Trends of Elemental Carbon and Char/Soot Ratios in Five Sediment Cores from Eastern China Marginal Seas: Indicators of Anthropogenic Activities and Transport Patterns. *Environmental Science & Technology* **52**, 9704-9712, doi:10.1021/acs.est.8b00033 (2018).

- 7 Masiello, C. A. New directions in black carbon organic geochemistry. *Marine Chemistry* **92**, 201-213, doi:<https://doi.org/10.1016/j.marchem.2004.06.043> (2004).
- 8 Hammes, K. *et al.* Comparison of quantification methods to measure fire-derived (black/elemental) carbon in soils and sediments using reference materials from soil, water, sediment and the atmosphere. *Global Biogeochemical Cycles* **21**, doi:<https://doi.org/10.1029/2006GB002914> (2007).
- 9 Han, Y. *et al.* Evaluation of the thermal/optical reflectance method for discrimination between char- and soot-EC. *Chemosphere* **69**, 569-574, doi:<https://doi.org/10.1016/j.chemosphere.2007.03.024> (2007).
- 10 Cong, Z. *et al.* Historical Trends of Atmospheric Black Carbon on Tibetan Plateau As Reconstructed from a 150-Year Lake Sediment Record. *Environmental Science & Technology* **47**, 2579-2586, doi:10.1021/es3048202 (2013).
- 11 Huang, C. *et al.* Anthropogenic-Driven Alterations in Black Carbon Sequestration and the Structure in a Deep Plateau Lake. *Environmental Science & Technology* **55**, 6467-6475, doi:10.1021/acs.est.1c00106 (2021).
- 12 Huang, C. *et al.* Spatial variation of particulate black carbon, and its sources in a large eutrophic urban lake in China. *Science of The Total Environment* **803**, 150057, doi:<https://doi.org/10.1016/j.scitotenv.2021.150057> (2022).
- 13 Han, Y. *et al.* Existence and Formation Pathways of High- and Low-Maturity Elemental Carbon from Solid Fuel Combustion by a Time-Resolved Study.

[General Comment 2]. The study measured and reported EFs from different sources (main types of BC emissions), but in developing inventories, ratios of char/soot were incorporated into a previously published inventory. Why not using directly measured EFs? By comparing EFs in the present study with literature studies, obvious differences were observed (R&D line and 100-105 and figure s1), and the authors explained this by different combustion and control technologies. Using the ratio in developing char- and soot-BC inventories is from the assumption of combustion processes and control technologies did not affect or change the ratio of char/soot, which may be not valid in most cases. This part needs to be clearly evaluated and probably updated.

[Response] Thank you for raising important issues. Regarding the question about the specific data selection criteria used for dividing the emission inventory, here is the detailed response:

Since the 1960s, China's economic situation and social policies have undergone tremendous changes. Taking diesel vehicles as an example, the black carbon emission factor of diesel cars has decreased by 3-4 orders of magnitude, from the previous emission standards to the recently widely promoted China VI standards¹⁴. Therefore, if we calculate the historical inventory using emission factors from our on-road tests (which currently include diesel vehicles ranging from China III to China VI standards), it would introduce significant errors and uncertainties, contradicting historical facts.

Based on these considerations, we choose not to use the emission factors from the latest tests (mainly after 2020) for inventory calculations but instead analyze the existing BC emission inventory using the $R_{C/S}$ ratio measured by us. However, it should be noted that, except for industrial sources, there have been no fundamental changes in combustion technology and combustion temperature for other emission sources. Therefore, we believe it is reasonable to analyze historical emission results using the current measured $R_{C/S}$ ratio. Of course, there is certainly some deviation between the current experimental results and historical data. At the same time, our existing assumptions definitely have some limitations. However, since we cannot access the historical $R_{C/S}$ ratio, the current assumption is the most reasonable solution under the existing conditions.

Additionally, in the latest version, we have redrawn Figure 2 and added uncertainty intervals based on the historical best estimate of $R_{CE/SE}$. These intervals can cover the deviation between the measured results and historical data. Here are the specific revised versions:

Fig. 2 Total BC and char to soot ratio of anthropogenic source emissions in China from 1960-2017 (a), temporal trends of annual emission of char and soot (b) (Shaded areas are uncertainty intervals of char and soot, represented by quartiles). The percentage contribution of each emission source to char (c) and soot (d) ("Other anthropogenic" refers to emissions from non-road transportation, "Energy sector" refers to emissions from power plants, specifically PCBs mentioned in the text.)

[General Comment 3] Emission factors are from laboratory and field-based measurements. It has been well recognized that results from these two approaches would be rather different, depending on methods and varying among fuel types and

combustion facilities in different sectors. The authors are suggested to comment on the differences on EFs from these two approaches, and how to integrate data from different methods for different sources, as well as its contribution to the inventory uncertainty.

[Response] Thank you very much for carefully reading our manuscript and raising professional questions. As you pointed out, there are significant differences between the results obtained from on-site measurements and those from laboratory simulation tests. This is because the combustion conditions and fuel properties during on-site measurements are typically uncontrollable, whereas the laboratory measurements are conducted under controlled fuel properties and combustion conditions. We have taken this factor into account in this article, and all emission inventory calculations are based on the results obtained from on-site measurements.

We also conducted extensive laboratory simulation work to provide alternative perspectives by setting up experiments with different stove combustion conditions and fuel types in a tubular furnace. The purpose of these experiments was to observe the differences in char and soot generation pathways under different combustion conditions and determine the influencing factors of their generation pathways.

In conclusion, in the last paragraph of the introduction, we supplemented a detailed inventory of samples from different emission sources and clarified the specific purposes of actual test samples and laboratory simulation samples. We also moved Table S5 to the manuscript of the article, allowing readers to have a more comprehensive understanding of our samples. Here are the specific revised versions:

Lines 82-87 in the revised Manuscript:

In this study, we collected combustion event data from three different sources, including 122 samples from RS, 160 samples from VE, and 12 samples from CB (specific information shown in Table 1). Based on this data, we established a comprehensive multi-sector database of char and soot emission factors. By further combining 105 sets of results from laboratory stove simulations, we identified key factors influencing the variation in char and soot emissions and explained the differences in their formation pathways.

[General Comment 4] The study suggested that future reductions should focus on char, in China and the world. Two main concerns here: with significant reductions of char-BC in China, why it would be still the main focus in future BC reduction but not the remaining part (question 1)? Secondly, without global inventories including its historical trends, it is too simplistic to conclude that the world BC reductions should focus on char based on the present analysis (question 2).

[Response] Thank you very much for providing professional question. In fact, in the implication section, we emphasized targeted combustion technology improvements for different emission sources. With the advancement of combustion technology, char primarily originates from low-temperature combustion emissions, making it a significant part of BC reduction and an area of focus for future emissions reduction efforts. With the continuous advancement of combustion technology, the emissions of char generated from low-temperature combustion will be significantly reduced. Therefore, in future BC emissions reduction, char will become a major component in

emissions reduction. In summary, we pay more attention to char in the historical context because it has been the main component of China's BC emissions for a long period of time. In future emission reduction policies, we also prioritize char because rapid reduction can be achieved by improving combustion technologies in various industries.

Regarding your second question, we have already mentioned in the results section the corresponding relationship between China's black carbon emissions from 1960 to 2017 and different levels of development worldwide. In the first phase (1960 to 1990), the contribution of black carbon (BC) emissions from different sources in China was similar to the results for developing countries in 2017. During this period, char emissions accounted for over 70% of the total BC emissions¹⁵. Additionally, according to Xu et al.'s global inventory of black carbon emissions, over 80% of BC emissions worldwide in 2017 came from developing countries. Taking these two points into consideration, we can estimate that at least 50% of global BC emissions come from char. Therefore, both in terms of global BC emissions contribution and the difficulty of reduction, char should receive more attention in future BC emission reduction efforts.

Regarding the global inventory and emission reduction policies for char and soot emissions, our co-author Professor Shen Huizhong has conducted detailed calculations based on global emission factors and historical data, and the manuscript will be submitted as soon as possible. In this article, our research focus is still within China, but we have made preliminary estimates of char and coal soot trends using measured data and historical inventories on a global scale. Due to significant differences between China and other countries in combustion technologies and emission standards, for the

sake of scientific rigor and accuracy, we have omitted specific emission reduction estimates in the final paragraph of the implications. Here are the specific revised versions:

Lines 308-317 in the revised Manuscript:

In fact, not only in China, but also globally, the combustion of low-temperature solid fuels and emissions from mobile sources contribute significantly to BC. For example, in 2017, in India, another developing country, the contribution of mobile sources and the combustion of low-temperature solid fuels to BC emissions was approximately 66%, whereas globally, this proportion reached as high as 77%¹⁵. Considering that most countries around the world are relatively lagging behind China in terms of combustion technology and emission standards, promoting these two technological improvements globally will greatly facilitate the reduction of BC emissions, especially in reducing char emissions primarily generated from low-temperature combustion.

15 Xu, H. *et al.* Updated Global Black Carbon Emissions from 1960 to 2017: Improvements, Trends, and Drivers. *Environmental Science & Technology* **55**, 7869-7879, doi:10.1021/acs.est.1c03117 (2021).

[General Comment 5] Are these direct evidences on the distinct light properties of char and soot-BC in literature studies (question 1)? some references cited in the introduction are not from the area of atmospheric science, in another words, they had

different definition and quantitation methods of char and soot in comparison with the present method (question 2).

[Response] Thank you very much for your insightful question. The first question will be one of the important goals for our future research. In fact, there is no research that clearly indicates their specific differences in climate change and health effects. This is because both char and soot are part of the continuum of brown-black carbon combustion products, which are simultaneously generated and difficult to separate during the combustion process. Due to the fact that in the IMPROVE protocol, char is quantified at a lower temperature compared to soot, it is challenging to separate char alone solely by improving existing methods. In previous studies, Han et al. and Corbin et al. observed significant differences between char and soot in terms of physical morphology, chemical bonding, and solubility using techniques such as transmission electron microscopy. These observation results indirectly validate the differences in their effects. Therefore, our current work mainly focuses on qualitatively distinguishing char and soot in the black carbon inventory, and future work will focus on separating pure char and quantitatively studying the differences between char and soot in terms of optical and toxicological properties.

Regarding your second question. The IMPROVE protocol and TOR method on OC/EC analyzers is now not only applied to atmospheric samples but also to BC samples collected in other domains such as the hydrosphere and pedosphere, The references we cited regarding char and soot in sediment and water bodies are quantitatively measured by the IMPROVE protocol and TOR method.

[Specific comment]

[Comment 1] Abstract – this part is suggested to be written. For example, the first two sentences could be shortened leaving more spaces for key results and findings. “a significant dataset”?

[Response] Thank you very much for providing insightful suggestions. We have made modifications to the abstract by shortening the description of the background section and enhancing the discussion in the conclusion part. The following is the revised content of the abstract:

Lines 27-39 in the revised manuscript:

Emission factors and inventories of black carbon (BC) aerosols are crucial for estimating their adverse atmospheric effect. However, it is imperative to separate BC emissions into char and soot subgroups due to their significantly different physicochemical properties and potential effects. Here, we present a substantial dataset of char and soot emission factors derived from field and laboratory measurements. Based on the latest results of the char-to-soot ratio, we further reconstructed the emission inventories of char and soot for the years 1960-2017 in China. Our findings indicate that char dominates annual BC emissions and its huge historical decrease, which can be attributable to the rapid changes in energy structure, combustion technology and emission standards in recent decades. Our results suggest that further BC emission reductions in both China and the world should focus on char, which mainly derives from lower-temperature combustion and is easier to decrease compared to soot.

[Comment 2] Lines 52-55, as mentioned, are direct studies on different physiochemical properties and especially light properties of char and soot-BC that is defined and analyzed by the same method as you did here? In other areas, “soot” or “char” being separated from combustion exhausts are rather different from those from the operational definition here.

[Response] Thank you for carefully reading our article. In Lines 52-55 in our manuscript, we discussed the differences in optical properties between char and soot and referenced two literature papers that quantitatively studied char and soot using the TOR method. Although these papers did not quantitatively describe the differences in optical properties between the two, they demonstrated from a structural perspective that their optical properties are distinct.

Besides, the IMPROVE protocol and TOR method on OC/EC analyzers is now not only applied to atmospheric samples but also to BC samples collected in other domains such as the hydrosphere and pedosphere, The references we cited regarding char and soot in sediment and water bodies are quantitatively measured by the IMPROVE protocol and TOR method.

[Comment 3] Line 58, “a single rule” or a single ratio? The rule is nearly the same and straightforward that is the ratio of char/soot for different sources, by fuels and combustion technologies, and the important issue here is specific and reliable (accurate) values of those ratios for different sources.

[Response] Your suggestions are very important for improving the readability and professionalism of our articles. The following are the revisions we have made to the manuscript:

Lines 57-60 in the revised manuscript:

However, separating BC into char and soot subgroups needs specific and reliable values of char to soot ratios for different emission sectors, resulting from differences in fuel composition, combustion, and after-treatment technologies.

[Comment 4] Line 65, which year? What about emissions from India, as the Chinese emissions are reducing?

[Response] Your suggestions have been very helpful in improving the scientific language expression of this manuscript. We have supplemented with the latest references and made the following modifications to the manuscript:

Lines 63-67 in the revised manuscript:

With the rapid development of the economy, China became the world's largest emitter of BC in 2000, accounting for about one-fourth of the global BC emissions¹⁶. However, with the update of air pollution control policies and optimization of combustion technologies, China's BC emissions decreased by about one-third by 2019¹⁷. The rapid change in China's BC emissions holds significant reference value for other developing countries.

[Comment 5] Line 80, in these 400 events (including repeats?), it is more necessary to

clearly show the sample sizes for each sector, and those from laboratory and field tests, and the representativeness of these events.

[Response] Thank you for your valuable suggestions. Following your advice, we have provided more detailed information about the samples in the last paragraph of the introduction section and moved Table S5 to the manuscript of the article, allowing readers to have a more comprehensive understanding of our samples.

Lines 82-87 in the revised manuscript:

In this study, we collected combustion event data from three different sources, including 122 samples from RS, 160 samples from VE, and 12 samples from CB (specific information shown in Table 1). Based on this data, we established a comprehensive multi-sector database of Char and Soot emission factors. By further combining 105 sets of results from laboratory stove simulations, we identified key factors influencing the variation in Char and Soot emissions and explained the differences in their formation pathways.

[Comment 6] Lines 105-108, the authors are suggested to carefully exam these data and well explain the differences observed. In figure s1, EFs from briquette coals were higher, and EF differences in brick kiln and coke oven were too large, in nearly 3-4 orders of magnitude. The authors stated many studies in past were prior to 2012, but in fact as seen in Table s2, there are few studies prior to 2012. The explanation of fuel composition and different after-treatment technology are too general and cannot well explain the difference.

[Response] Thank you for your very professional advice. This suggestion significantly enhances the professionalism and rigor of our manuscript. In the manuscript, we want to clarify that when we mention "many previous studies conducted before 2012," it does not include the recent BC measurement data we collected in Table S2. Instead, it refers to the measurement data used by Xu et al. when calculating the Chinese BC emission inventory from 1960 to 2017. In summary, our findings indicate that the BC emission factor for domestic stoves has remained relatively unchanged, while the BC emission factors for road and non-road vehicles have significantly decreased due to updated emission standards. Furthermore, the BC emission factor for industrial sources has also decreased significantly due to the closure of domestic coking ovens and the implementation of standardized emission regulations. To prevent any misunderstanding among readers, we have made the following modifications in the manuscript:

Lines 108-116 in the revised manuscript:

Compared to the EF_{BC} data used in the historical inventory (Fig. S1), we observe the following changes: For RS, the EF_{BC} for RBB has not significantly changed, while the EF_{BC} for RCC has decreased by about 50% due to continuous improvements in stove technology. The EF_{BC} for VE emissions has decreased by one to two orders of magnitude due to updated emission standards and fuel quality. For CB, especially PCB, the EF_{BC} has decreased by 3-4 orders of magnitude, attributed to the closure of residential coke ovens and the continuous strengthening of emission standards. Moreover, the trend of the EF_{BC} in our results is consistent with the data found in the literature within the past five years (Table S2).

[Comment 7] Lines 110-120, please add uncertainty range of these numbers.

[Response] Thank you for your suggestion. We have conducted a comprehensive check on the results section to ensure that each reported average value is accompanied by the corresponding uncertainty.

[Comment 8] Line 123, how to ensure results are “more accurate”?

[Response] Thank you for your careful review. We have used incorrect adjectives in the manuscript, and the specific revisions are as follows:

Lines 130-131 in the revised manuscript:

The ratio of EF_{char} to EF_{soot} ($R_{C/S}$) was calculated to gain a more intuitive view on the contribution of these two subgroups to the EF_{BC} across various sectors.

[Comment 9] Lines 141-142, rethink this assumption

[Response] Thank you very much for your suggestions. Regarding the issue of assumption credibility, we have provided a detailed explanation and discussion in response to the second general comment.

[Comment 10] Lines 150 -170, are BC emissions reported here are from the present calculation, or the paper of Xu et al.?

[Response] You have raised a key question. All the data in the results section regarding annual BC emissions were sourced from Xu et al.'s paper, and we further disaggregated

this total amount. We have added corresponding references for all the cited parts in the manuscript.

[Comment 11] Line 173, may be helpful to add the reduction percentages, and variations in these estimates.

[Response] Your suggestion allows the reader to further visualize the variability of the data. We have made the following revises in the manuscript:

Line 180-182 in the revised manuscript:

During the PIII period, annual BC emissions saw a gradual decline from 1538 Gg in 2001 to 1229 Gg in 2017, representing a reduction of approximately 20%. However, the trends for charcoal and soot emissions differed. Charcoal emissions experienced a rapid decrease from 890 Gg to 625 Gg, reflecting a decline of nearly 40%, whereas soot emissions only slightly dropped from 648 Gg to 604 Gg, indicating a decrease of less than 7%.

[Comment 12] Line 176-180, it is not very clear in the logistic relationship between the study cited here and the discussion in this part. The value was 1.41 (despite of different analytical methods and term definition) in 2019, that was close to the 1.37 in the 2017 in your inventory.

[Response] Your question prompts us to further examine our own article from an academic perspective. In fact, the $R_{CE/SE}$ in the emission inventory listed in the results section is calculated based on the RC/S of each emission source and the activity level

of each emission source in that year. In other words, it is only a theoretical value. On the other hand, the $R_{C/S}$ in the surface water of Lake Taihu (determined using Improve-A-TOR method) reflects the actual $R_{CE/SE}$ values dissolved into the water through sedimentation in that year. We found that the theoretical values are very close to the actual values, which provides good support for the validity of our inventory.

[Comment 13] Lines 201-212, since difference methods and even definition of terms here, how to be sure these results, evening the trends, can be compared and mutually corroborated? It is suggested to comment on the uncertainty and cautions in exploring the results here.

[Response] Thank you for your careful review of our article. The IMPROVE protocol and TOR method on OC/EC analyzers is now not only applied to atmospheric samples but also to BC samples collected in other domains such as the hydrosphere and pedosphere, The references we cited regarding char and soot in sediment and water bodies are quantitatively measured by the IMPROVE protocol and TOR method.

[Comment 14] Lines 229-273, suggest to shorten and be more specific as some were discussed in previous publications resulting in limited new information here (and some still needs more experimental and modelling evidences to valid the mechanisms).

[Response] Thank you very much for providing professional advice. Actually, our research group has been very interested in the formation mechanisms of char and Soot. In the early versions of this article, we included this part as a core section. However,

due to many unknown aspects that require further experimentation to explore, we decided to simplify this section and plan to discuss it in detail in another future article. Nevertheless, this part in the main text is still very important as it expands our understanding of existing formation pathways and identifies several key features of Char formation based on limited experimental data. Based on your suggestions, we have condensed the discussion on the mechanism section to highlight the main content of the article. Here are the specific modifications we made in the article:

Line 237-266 in the revised manuscript:

In the past few decades, through experimental, theoretical, and computational work, many pathways for the formation of soot have been proposed¹⁸⁻²⁰. However, the specific mechanisms of char formation have not been clearly elucidated to date. Through controlled combustion experiments, we have observed limited information, revealing differences between char and coal smoke in terms of precursors and byproducts.

Firstly, compared to soot, char may be formed from higher molecular weight organic volatile precursors. We observed that the first stage of coal combustion produces smaller organic molecules, rapidly generating coal smoke, while the second stage of pyrolysis produces larger organic fragments or tar, accompanied by the rapid formation of char²¹. This phenomenon may also occur in internal combustion engines, where lower molecular weight components rapidly evaporate when the fuel mixture is injected into compressed hot air, resulting in the combustion of soot upon contact with oxygen, while higher molecular weight components form char due to limited contact

with oxygen^{22,23}. Our experimental data confirms this inference: under low engine load or idle mode, the $R_{C/S}$ is higher, while under high engine load or operational mode, the $R_{C/S}$ is lower (refer to Table S1).

Secondly, different by-products or intermediate products generate between char and soot formation pathways. We found that the high time-resolved $R_{C/S}$ were closely proportional to higher ratios of high-molecular-weight (4-6 ring) to low-molecular-weight (2-3 ring) PAHs as well as higher ratios of refractory organic carbon to volatile organic carbon of OC fractions^{13,24}. Furthermore, oxygenated polycyclic aromatic compounds (oPACs) are likely to be related to char production compared to parent PAHs in traditional soot formation process. This is supported by the fact that the oPACs content of RS samples (with high $R_{C/S}$) is three times higher than that of VE samples (with relatively low $R_{C/S}$)²⁵. The oPACs are believed to derive from the conversion of oxygen-containing functional groups in the fuel during combustion²⁶. Therefore, we can preliminarily conclude that that when fuels with relatively complex compositions (especially those with higher oxygen content) are burning at lower temperatures (Fig. 3b), more char than soot will be produced.

13 Han, Y. *et al.* Existence and Formation Pathways of High- and Low-Maturity Elemental Carbon from Solid Fuel Combustion by a Time-Resolved Study. *Environmental Science & Technology* **56**, 2551-2561, doi:10.1021/acs.est.1c05216 (2022).

18 Frenklach, M. & Wang, H. Detailed modeling of soot particle nucleation and

- growth. *Symposium (International) on Combustion* **23**, 1559-1566, doi:[https://doi.org/10.1016/S0082-0784\(06\)80426-1](https://doi.org/10.1016/S0082-0784(06)80426-1) (1991).
- 19 Shukla, B. & Koshi, M. A highly efficient growth mechanism of polycyclic aromatic hydrocarbons. *Physical Chemistry Chemical Physics* **12**, 2427-2437, doi:10.1039/B919644G (2010).
- 20 Kholghy, M. R., Kelesidis, G. A. & Pratsinis, S. E. Reactive polycyclic aromatic hydrocarbon dimerization drives soot nucleation. *Physical Chemistry Chemical Physics* **20**, 10926-10938, doi:10.1039/C7CP07803J (2018).
- 21 Han, Y. *et al.* High Time- and Size-Resolved Measurements of PM and Chemical Composition from Coal Combustion: Implications for the EC Formation Process. *Environmental Science & Technology* **52**, 6676-6685, doi:10.1021/acs.est.7b05786 (2018).
- 22 Kayes, D. & Hochgreb, S. Mechanisms of Particulate Matter Formation in Spark-Ignition Engines. 1. Effect of Engine Operating Conditions. *Environmental Science & Technology* **33**, 3957-3967, doi:10.1021/es9810991 (1999).
- 23 Kayes, D. & Hochgreb, S. Mechanisms of Particulate Matter Formation in Spark-Ignition Engines. 3. Model of PM Formation. *Environmental Science & Technology* **33**, 3978-3992, doi:10.1021/es981101o (1999).
- 24 Wang, J. *et al.* Emission characteristics and influencing mechanisms of PAHs and EC from the combustion of three components (cellulose, hemicellulose, lignin) of biomasses. *Science of The Total Environment* **859**, 160359,

doi:<https://doi.org/10.1016/j.scitotenv.2022.160359> (2023).

- 25 Han, Y. & Chen, Y. Aromatic components of brownness carbon in the formation of low-maturity elemental carbon from combustion of Biomass, Coal, and diesel by a time resolved Fourier transform ion cyclotron resonance mass spectrometry (FT-ICR MS) study. *Environmental Science & Technology* **In submission** (2023).
- 26 Krzyszczyk, A. & Czech, B. Occurrence and toxicity of polycyclic aromatic hydrocarbons derivatives in environmental matrices. *Science of The Total Environment* **788**, 147738, doi:<https://doi.org/10.1016/j.scitotenv.2021.147738> (2021).

[Comment 15] Line 310, if solid fuels were not burned in stoves, which energy would be used in the residential sector? Will that produce relatively more soot-BC if modern energies like gas fuels and electricity from coal-fired plants were used?

[Response] Thank you very much for your careful reading of our article and for asking insightful questions. In the article, we have mentioned that the best approach would be to use industrial boilers to burn all solid fuels and deliver energy to residential areas, or to significantly upgrade domestic combustion furnaces to meet industrial emission standards. In this case, since char is the easier part to reduce emissions, the proportion of Soot in black carbon may increase. However, in terms of absolute emission levels, both char and soot emissions will certainly be significantly reduced.

[**Comment 16**] Lines 310-315, do you mean also burn solid fuels in boilers in India and enforcing vehicle emissions in US? BC sources, as well as EFs in other countries could be very different from that in China, and vehicle emissions were affected by not only fuel types but road conditions that are very different between China and US. How about the accuracy in the estimated numbers like 64%, 75%, and 53%? The calculation method and future reduction scenarios of BC emission from India and US is not available in the Method section.

[**Response**] Your feedback has further prompted us to reflect on the validity of the calculations in the manuscript. In the previous calculations, we referred to Xu et al.'s emissions inventory to obtain the annual BC emissions of other countries. We combined this data with China's emission factor and reduction policies to calculate possible future emission scenarios. However, as you pointed out, we lack actual measurements to support the corresponding emission factors in other countries' source profiles, especially for vehicles with different emission standards and other types of furnaces used in Europe and America. This significantly affects the uncertainty of the final calculation results. Regarding the global inventory and emission reduction policies for char and soot emissions, our co-author Professor Shen Huizhong has conducted detailed calculations based on global emission factors and historical data, and the manuscript will be submitted as soon as possible. In this article, our research focus is still within China, but we have made preliminary estimates of char and coal soot trends using measured data and historical inventories on a global scale. Due to significant differences between China and other countries in combustion technologies and

emission standards, for the sake of scientific rigor and accuracy, we have omitted specific emission reduction estimates in the final paragraph of the implications. Here are the specific revised versions:

Lines 308-317 in the revised Manuscript:

In fact, not only in China, but also globally, the combustion of low-temperature solid fuels and emissions from mobile sources contribute significantly to BC. For example, in 2017, in India, another developing country, the contribution of mobile sources and the combustion of low-temperature solid fuels to BC emissions was approximately 66%, whereas globally, this proportion reached as high as 77%¹⁵. Considering that most countries around the world are relatively lagging behind China in terms of combustion technology and emission standards, promoting these two technological improvements globally will greatly facilitate the reduction of BC emissions, especially in reducing char emissions primarily generated from low-temperature combustion.

15 Xu, H. *et al.* Updated Global Black Carbon Emissions from 1960 to 2017: Improvements, Trends, and Drivers. *Environmental Science & Technology* **55**, 7869-7879, doi:10.1021/acs.est.1c03117 (2021).

[Comment 17] Line 372, what do you mean “timeliness” here?

[Response] Thank you for your question. In our response to general comment 2, we discussed in detail the issue of different historical periods possibly corresponding to

different historical emission sources and BC emission factors.

[Comment 18] Line 372-376, in your present study or Xu et al.?

[Response] Thank you for your question. The timeliness and computational methods are areas of our research, and for the specific implementation of Monte Carlo simulation, we have referenced the work of Xu et al¹⁵.

[Comment 19] Line 379 and 381, which parameters? Please be more specific

[Response] Thank you for your careful reading of our article. We referenced the study by Xu et al¹⁵. for the simulation and calculation section of our paper on the historical inventory of black carbon. The specific required parameters can be found in their article.

[Comment 20] Figure s1, what are the triangles in the figure? What about variations in these EFs that are usually large? is 'Historical EF' the "previous inventory" in figure caption? "measured data"- "updated inventory"? inventory or only EFs ?

[Response] Thank you very much for your suggestion. To make the part of BC emission factor close to zero in the Fig. S1 clearly visible, we have added triangular symbols at the top of each bar chart. We have made revisions to the misleading descriptions in the legend to make them clearer and more precise. Meanwhile, we provided further description and explanation of the diversity shown in Figure S1 in the manuscript.

Lines 108-116 in the revised manuscript:

Compared to the EF_{BC} data used in the historical inventory (Fig. S1), we observe

the following changes: For RS, the EF_{BC} for RBB has not significantly changed, while the EF_{BC} for RCC has decreased by about 50% due to continuous improvements in stove technology. The EF_{BC} for VE emissions has decreased by one to two orders of magnitude due to updated emission standards and fuel quality. For CB, especially PCB, the EF_{BC} has decreased by 3-4 orders of magnitude, attributed to the closure of residential coke ovens and the continuous strengthening of emission standards. Moreover, the trend of the EF_{BC} in our results is consistent with the data found in the literature within the past five years (Table S2).

Figure S1. Historical EF_{BC} ^{15,27} data vs. measured data

15 Xu, H. *et al.* Updated Global Black Carbon Emissions from 1960 to 2017: Improvements, Trends, and Drivers. *Environmental Science & Technology* **55**,

7869-7879, doi:10.1021/acs.est.1c03117 (2021).

- 27 Wang, R. *et al.* Trend in Global Black Carbon Emissions from 1960 to 2007. *Environmental Science & Technology* **48**, 6780-6787, doi:10.1021/es5021422 (2014).

Reviewer #1 (Remarks to the Author):

The authors have addressed all of my concerns. I have no further comments and suggestions.

Reviewer #3 (Remarks to the Author):

this revised version is clear and clarified in main concerns. I think the separation of char- and soot-EC would be valuable in future assessing climate impacts of carbonaceous aerosols